# Allosteric activation of the SPRTN protease by ubiquitin maintains genome stability

Sophie Dürauer[1,2], Hyun-Seo Kang [3,4], Christian Wiebeler [5], Yuka Machida[6], Dina S. Schnapka[1,2], Denitsa Yaneva[1,2], Christian Renz [7], Maximilian J. Götz[1,2], Pedro Weickert[1,2], Abigail C. Major[5], Aldwin S. Rahmanto[7,8], Sophie M. Gutenthaler-Tietze [9,10], Lena J. Daumann [9], Petra Beli[7,8], Helle D. Ulrich [7], Michael Sattler[3,4], Yuichi J. Machida [6], Nadine Schwierz[5] & Julian Stingele [1,2] ✉

The DNA-dependent protease SPRTN maintains genome stability by degrading toxic DNA-protein crosslinks (DPCs). To understand how SPRTN's promiscuous protease activity is confined to cleavage of crosslinked proteins, we reconstitute the repair of DPCs including their modification with SUMO and ubiquitin chains in vitro. We discover that DPC ubiquitylation strongly activates SPRTN independently of SPRTN's known ubiquitin-binding domains. Using protein structure prediction, MD simulations and NMR spectroscopy we reveal that ubiquitin binds to SPRTN's protease domain, promoting an open, active conformation. Replacing key interfacial residues prevents allosteric activation of SPRTN by ubiquitin, leading to genomic instability and cell cycle defects in cells expressing truncated SPRTN variants that cause premature aging and liver cancer in Ruijs-Aalfs syndrome patients. Collectively, our results reveal a ubiquitin-dependent regulatory mechanism that ensures SPRTN activity is deployed precisely when and where it is needed.

Cells invest in extensive repair mechanisms to ensure fidelity of the genetic information stored in their DNA. Defective DNA repair results in mutagenesis and genome instability, major hallmarks of cancer, aging and aging-related diseases[1,2]. Cellular DNA repair activities are organized by sophisticated networks of post-translational modifications[3,4]. Regulatory ubiquitylation events are critical to recruit DNA repair factors in highly controlled manners. Mono-ubiquitylation of PCNA promotes DNA damage tolerance by recruiting translesion synthesis (TLS) polymerases[5], while mono-ubiquitylation of the FANCD2/FANCI heterodimer traps the complex on DNA, initiating DNA repair by the Fanconi anemia pathway[6].

Tight regulation is especially important for DNA repair enzymes that are potentially toxic. The SPRTN protease employs a promiscuous activity to degrade covalent DNA-protein crosslinks (DPCs), but it has remained enigmatic how the enzyme achieves specificity for crosslinked proteins and how the unwanted cleavage of chromatin proteins is prevented. DPCs arise upon stabilization of covalent intermediates between DNA-processing enzymes and their substrates[7]. Additionally, various endogenous and environmental reactive agents crosslink proteins to DNA[8,9]. DPCs are toxic because they block DNA replication and transcription[10–13]. The collision of the replication machinery with crosslinked proteins initiates repair by SPRTN[14,15], which can

[1]Gene Center, Ludwig-Maximilians-Universität München, Munich, Germany. [2]Department of Biochemistry, Ludwig-Maximilians-Universität München, Munich, Germany. [3]Institute of Structural Biology, Molecular Targets and Therapeutics Center, Helmholtz Munich, Neuherberg, Germany. [4]Bavarian NMR Center and Department of Bioscience, TUM School of Natural Sciences, Technical University of Munich, Garching, Germany. [5]Institute of Physics, University of Augsburg, Augsburg, Germany. [6]Developmental Therapeutics Branch, Center for Cancer Research, National Cancer Institute, Bethesda, MD, USA. [7]Institute of Molecular Biology gGmbH, Mainz, Germany. [8]Institute of Developmental Biology and Neurobiology (IDN), Johannes Gutenberg-Universität Mainz, Mainz, Germany. [9]Chair of Bioinorganic Chemistry, Heinrich-Heine Universität Düsseldorf, Düsseldorf, Germany. [10]Department of Chemistry, Ludwig-Maximilians-Universität München, Munich, Germany. ✉e-mail: stingele@genzentrum.lmu.de

additionally be triggered by global-genome mechanisms[9]. The repair of DPCs by SPRTN is essential for viability. Its loss is lethal in human cell lines[16] and leads to dramatic genome instability and early embryonic lethality in mice[17].

SPRTN features a metalloprotease domain at the *N*-terminus, which, together with the single-stranded DNA (ssDNA) -binding zinc-binding domain (ZBD), forms the conserved SprT domain (Fig. 1a)[18,19]. The SprT domain is followed by a basic region (BR) that interacts with double-stranded DNA (dsDNA)[20]. ZBD and BR couple SPRTN activity to the recognition of ssDNA-dsDNA junctions[21], that arise when DNA polymerases stall at DPCs during replication[14]. However,

the recognition of DNA junctions cannot explain how specificity is achieved during DPC repair, given that these structures are common throughout the genome, for example on the lagging strand during DNA replication. In addition to its DNA-binding domains, SPRTN bears interaction motifs for binding to the segregase p97 (SHP box) and PCNA (PIP box)[22–25] but neither is required for SPRTN's DPC repair function[9,14,17]. Furthermore, SPRTN carries a *C*-terminal ubiquitin-binding zinc finger (UBZ), promoting SPRTN ubiquitylation and thereby its inactivation[26]. A motif interacting with ubiquitin (MIU) has been predicted at SPRTN's *N*-terminus but has not been experimentally confirmed[27]. The presence of ubiquitin-binding

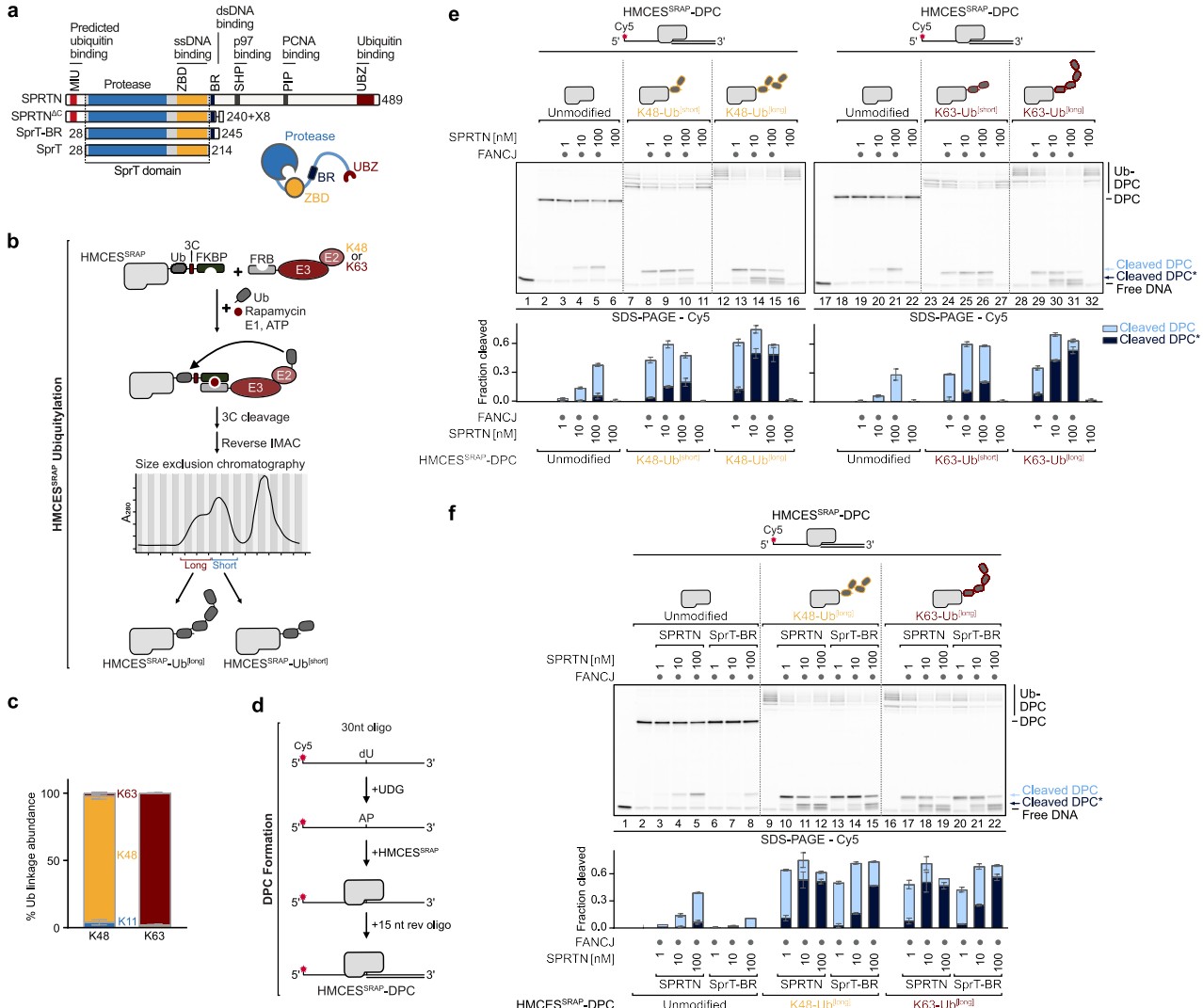

**Fig. 1 | Ubiquitylation of DPCs promotes their cleavage by SPRTN. a** Schematic of SPRTN's domain structure and truncated variants, featuring motif interacting with ubiquitin (MIU), protease domain, zinc-binding domain (ZBD), basic region (BR), SHP box for p97-binding, PCNA-interacting motif (PIP) and ubiquitin-binding zinc finger (UBZ). SPRTN^ΔC is caused by a frameshift mutation resulting in a variant composed of SPRTN's *N*-terminal 240 residues followed by eight additional amino acids (X8). **b** Schematic of HMCES^SRAP ubiquitylation to generate DPCs shown in **e**, **f**, Fig. 4 and Supplementary Fig. 5b and 6b. HMCES^SRAP-Ub(G76V)-3C-FKBP was incubated with FRB-E3 + E2 (K48 or K63) in the presence of ubiquitin, rapamycin, ubiquitin-E1 and ATP for 2 h (K63) or 6.5 h (K48) at 30 °C. After cleavage of the FKBP-tag via 3C-protease, ubiquitylated HMCES^SRAP was purified by reverse immobilized metal affinity chromatography (IMAC) and size-exclusion chromatography (SEC). **c** Mass spectrometry analysis of ubiquitin linkages formed by ubiquitylation of HMCES^SRAP as shown in (**b**). Bar chart shows the mean ± SD of three biological replicates. **d** Schematic of the generation of HMCES^SRAP-DPCs. HMCES^SRAP was

incubated for 30 min at 37 °C with a Cy5-labeled 30nt oligonucleotide containing a dU at position 15 and UDG. After crosslinking a complementary 15nt reverse oligonucleotide was annealed to form a ssDNA-dsDNA junction. **e** Indicated HMCES^SRAP-DPCs (10 nM) were incubated alone or in the presence of FANCJ (100 nM) and indicated concentrations of SPRTN (1-100 nM) for 1 h at 30 °C. Quantification: bar graphs represent the mean ± SD of three independent experiments. All samples derive from the same experiment and gels were processed in parallel. Values for cleavage of unmodified HMCES^SRAP-DPC are the same as in Supplementary Fig. 1b. Source data are provided as a Source Data file. **f** Indicated HMCES^SRAP-DPCs (10 nM) were incubated alone or in the presence of FANCJ (100 nM) and indicated concentrations of SPRTN or SprT-BR (1-100 nM) for 1 h at 30 °C. Quantification: bar graphs represent the mean ± SD of three independent experiments. All samples derive from the same experiment and gels were processed in parallel. Source data are provided as a Source Data file.

domains indicates a critical role of ubiquitin in regulating SPRTN-mediated DPC repair.

Indeed, DPCs are ubiquitylated during replication by the ubiquitin-E3s TRAIP and RFWD3[14,15,28], while SUMOylation precedes ubiquitylation of the protein adduct by the SUMO-targeted ubiquitin-E3s RNF4 and TOPORS during global-genome repair[9,29–32]. DPC ubiquitylation can promote proteasomal degradation of crosslinked proteins[9,14,15,29,30], but it has remained controversial whether it is important for SPRTN-mediated repair. Cleavage of a model DPC by SPRTN in frog egg extracts occurs even if the protein adduct has been treated with formaldehyde to prevent ubiquitylation[14]. Nonetheless, ubiquitylated DPCs accumulate upon SPRTN depletion[33], indicating that they are substrates of the protease. Furthermore, SPRTN's UBZ domain supports efficient DPC cleavage in frog egg extracts and cells[9,14], which has led to the speculation that the UBZ may help to recruit SPRTN to ubiquitylated DPCs. Surprisingly however, the UBZ domain is not essential for SPRTN function. Patients with Ruijs-Aalfs syndrome (RJALS) express truncated versions of SPRTN that lack the *C*-terminal part of the enzyme including the UBZ (SPRTN[ΔC], Fig. 1a)[27]. RJALS patients suffer from premature aging and liver cancer[27], phenotypes that are recapitulated in mice with reduced SPRTN function[17]. Yet, truncated SPRTN patient variants are clearly compatible with life, in contrast to full loss of SPRTN. Indeed, the severe growth defects associated with SPRTN loss in conditional mouse knock-out cells are rescued by expression of a truncated SPRTN variant[34]. It has remained enigmatic how SPRTN patient variants target DPCs in the absence of the UBZ and, more generally, whether and how SPRTN activity is regulated by DPC ubiquitylation.

Here, we investigate the role of ubiquitin in SPRTN activation by biochemical reconstitution of DPC ubiquitylation, molecular dynamics (MD) simulations, NMR experiments and cellular assays. We find that DPC ubiquitylation activates SPRTN more than one hundred-fold. Activation occurs independently of SPRTN's UBZ domain but involves a ubiquitin-binding interface at the back of its protease domain. This interface is required in cells expressing truncated RJALS patient variants to maintain genome stability and cellular fitness. Collectively, our results reveal a regulatory mechanism that confines SPRTN's protease activity by linking its activation to DPC modification. Moreover, given that ubiquitin-dependent activation is retained in truncated SPRTN variants, our data explain how residual SPRTN function is maintained in RJALS patients.

## Results
### Ubiquitylation of DNA-protein crosslinks promotes their cleavage by SPRTN
To directly test whether DPC ubiquitylation regulates SPRTN, we reconstituted DPC ubiquitylation in vitro. To modify DPCs with ubiquitin chains of defined linkages, we employed synthetic engineered ubiquitin-E3s (streamlined versions of the previously described Ubiquiton system[35]), enabling us to modify the catalytic SRAP domain of HMCES (HMCES[SRAP]) with K48- or K63-linked ubiquitin chains prior to DPC formation with an oligonucleotide containing an abasic (AP) site. HMCES actively crosslinks to AP sites within ssDNA to prevent AP site scission during DNA replication[36]. First, we fused a *C*-terminal tag containing a mono-ubiquitin moiety and a FK506-binding protein (FKBP) domain to HMCES[SRAP]. We then incubated this substrate with ubiquitin, an engineered ubiquitin-E3 carrying an FKBP-rapamycin-binding (FRB) domain, ubiquitin-E1, ubiquitin-E2, ATP and rapamycin (Fig. 1b). Rapamycin induces proximity between the substrate and the E3, promoting modification of the ubiquitin moiety fused to HMCES[SRAP] with either K48- or K63-linked polyubiquitin chains (depending on the identity of the E2/E3 enzymes used in the assay). Following cleavage of the 3C-site between ubiquitin and FKBP, HMCES[SRAP] modified with short or long ubiquitin chains was purified over several steps (Fig. 1b and Supplementary Fig. 1a, for all

recombinant proteins used in this study). Mass spectrometry (MS) analysis confirmed the specific formation of K48- and K63-linked polyubiquitin chains on HMCES[SRAP] (Fig. 1c). DPCs were then generated by incubating unmodified or ubiquitylated HMCES[SRAP] with an AP site-containing fluorescently-labeled ssDNA-dsDNA junction (Fig. 1d)[37,38].

Next, we incubated the DPCs with SPRTN and the helicase FANCJ, which is required for SPRTN activity in these assays. FANCJ loads on the ssDNA portion of the substrate and translocates into the crosslinked protein, resulting in unfolding of the protein adduct, which in turn enables SPRTN to cleave the DPC[37]. SPRTN cleaved ubiquitylated DPCs more efficiently than unmodified protein adducts, with long chains activating stronger than shorter ones, independently of linkage type (Fig. 1e, lanes 7-16 (K48) and lanes 23-32 (K63)). The ubiquitin-dependent activation of SPRTN was substantial with the extent of cleavage of ubiquitylated DPCs by 1 nM of SPRTN being comparable to the cleavage of unmodified DPCs by 100 nM of SPRTN (Fig. 1e, compare lanes 5 and 13 (K48) and lanes 21 and 29 (K63)). Remarkably, in addition to the fragment produced upon cleavage of unmodified DPCs (Fig. 1e, Cleaved DPC), smaller cleavage products (Fig. 1e, Cleaved DPC*) appeared upon cleavage of ubiquitylated DPCs. Of note, smaller cleavage products were also detected upon addition of free K48- or K63-linked tetra-ubiquitin chains, although to a lesser extent (Supplementary Fig. 1b, cleaved DPC*, lanes 7-9 (K48) and lanes 17-19 (K63)).

To test whether SPRTN's known ubiquitin-binding domains are mediating the stimulating effect of DPC ubiquitylation, we utilized a minimal active SPRTN variant (SprT-BR, aa28-245), that lacks both, MIU and UBZ (Fig. 1a). While the truncated SprT-BR variant showed reduced cleavage of unmodified DPCs compared to the wild-type (WT) enzyme (Fig. 1f, compare lanes 3-5 with lanes 6-8), DPC ubiquitylation strongly boosted its activity (Fig. 1f, compare lanes 10-12 with lanes 13-15 (K48) and lanes 17-19 with lanes 20-22 (K63)). The stimulating effect of DPC ubiquitylation on truncated SprT-BR suggested to us that this region likely contains an additional ubiquitin-binding site that mediates the effect of ubiquitin on SPRTN activation.

### Ubiquitin promotes an open SPRTN conformation
To explore this possibility, we used ColabFold[39] to predict complexes between SprT-BR and ubiquitin. In the top-ranked model, the hydrophobic Ile44-patch of ubiquitin was predicted to interact with a hydrophobic interface at the back of the SprT domain (Supplementary Fig. 2a-b), hereafter referred to as ubiquitin-binding interface at the SprT domain (USD). Interestingly, in all models, the SprT domain was predicted to adopt an open conformation with a highly accessible active site facing the DNA binding site of the ZBD. A similar conformation was also predicted in the absence of ubiquitin, in stark contrast to the published crystal structure of the SprT domain (PDB:6mdx[19]) that shows a closed conformation with the ZBD restricting access to the active site (Fig. 2a–c).

To explore whether the predicted open SprT conformation is in equilibrium with the closed conformation and whether ubiquitin binding may affect SprT conformation, we conducted all-atoms MD simulations. We used either the crystal structure or ColabFold-based predictions of the SprT domain, alone or in combination with ubiquitin, as starting points (Fig. 2d–f and Supplementary Fig. 2c). The compact conformation observed in the crystal structure remained largely unchanged over the entire 400 ns timeframe in three independent simulations (Fig. 2d and Supplementary Movie 1). To reveal the predominant conformations within all simulations, we employed RMSD-based clustering (Fig. 2g-i), revealing a single cluster with a closed conformation (Fig. 2g). In contrast, simulations of the ColabFold-predicted SprT structure displayed larger conformational changes during the simulations (Fig. 2e). We observed collapses to a compact conformation with a smaller radius of gyration (Fig. 2e, red arrow). Collapses were followed by rapid reopening of the structure

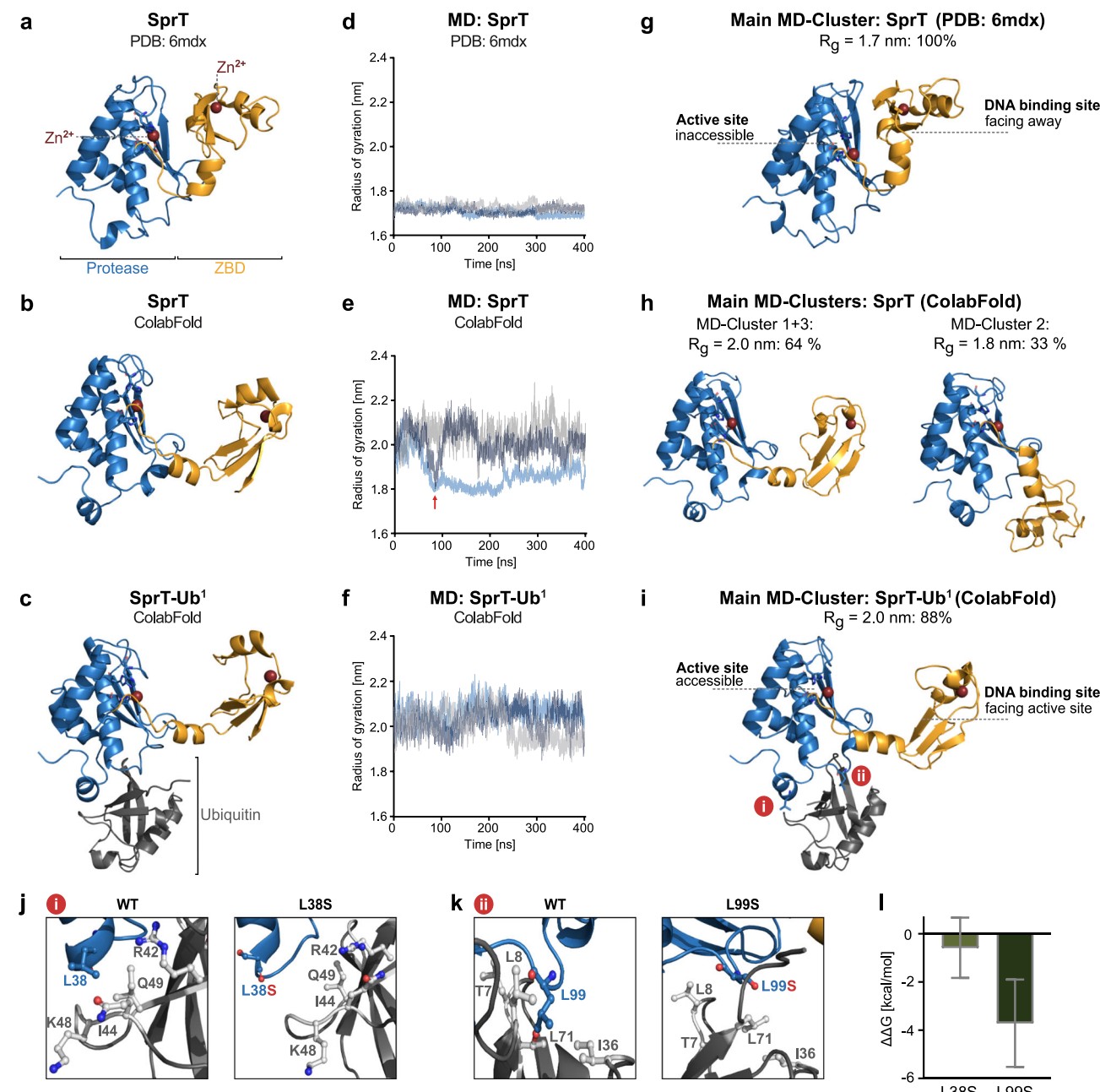

**Fig. 2 | Ubiquitin promotes an open SPRTN conformation. a–c** Experimental structure of SPRTN's SprT domain (SPRTN[aa28-214]), PDB: 6mdx (**a**), ColabFold predicted structure of SprT (**b**) and ColabFold predicted structure of a SprT-ubiquitin (Ub[1]) complex (**c**). Protease domain is colored in blue, zinc-binding domain (ZBD) in orange and the Ub[1] in grey. $Zn^{2+}$ ions are colored in red. **d–f** Radius of gyration (Rg) of the indicated structures over 400 ns of molecular dynamics (MD) simulation. Each curve represents an independent MD trajectory (*n* = 3). Source data are provided as a Source Data file. **g–i** Main MD-clusters of the indicated structures during MD simulation for 400 ns, generated from three independent trajectories. For SprT (ColabFold predicted) two of three main MD-clusters are depicted. Rg correlating frequencies among all performed simulations are labeled above the structures. **j, k** Zoom-in to regions **i** and **ii** of the SprT-Ub[1] complex (**i**), showing amino acids of ubiquitin (in grey) surrounding residue Leu38 (**j**) or L99 (**k**) of SPRTN (in blue) in the wild-type (WT) protein (left) and upon L38S or L99S replacement, respectively (right). **l** SprT-Ub[1] binding energy difference (ΔΔG) between SprT-L38S or -L99S and WT protein obtained from alanine scanning. Bar graphs show the mean ± SD of 301 snapshots from PBSA calculations for the central structure of the largest cluster. Source data are provided as a Source Data file.

(Fig. 2e, dark blue trace) or retention of the compact conformation (Fig. 2e, light blue trace, and Supplementary Movie 2). Clustering revealed two clusters with an open conformation (Fig. 2h, left) and one cluster with a closed conformation (Fig. 2h, right). Strikingly, the presence of ubiquitin prevented transitions of the SprT domain to the closed conformation (Fig. 2f and Supplementary Movie 3) and simulations predominantly remained in an open conformation (Fig. 2i). Moreover, ubiquitin binding to the USD interface of the SprT remained

stable across all three independent simulations (Fig. 2f). These data indicated to us that ubiquitin binding at the SprT domain may promote SPRTN activation by stabilizing an open conformation of the enzyme with an accessible active site.

Next, we wanted to determine amino acid residues within the USD interface that are important for ubiquitin-binding. In the predicted SprT-ubiquitin complex, Leu38 and Leu99 of SPRTN appeared to mediate the interaction via hydrophobic interactions involving

multiple amino acids within ubiquitin's hydrophobic Ile44- and Ile36-patch, respectively (Fig. 2i-k and Supplementary Fig. 2d-e). Both residues, Leu38 and Leu99, are highly conserved throughout evolution (Supplementary Fig. 2f). To assess the effect of replacing either leucine residue with a hydrophilic serine (L38S, L99S), we conducted free energy end-point calculations using MMPBSA in conjunction with alanine scanning (see Methods for details), which enabled us to quantify the effect of each leucine-to-serine replacement to the overall binding affinity of the SprT-ubiquitin complex. We calculated a decrease in binding affinity of around 0.6 kcal/mol for the L38S replacement and a more substantial decrease of 3.74 kcal/mol for L99S (Fig. 2l). This effect is explained by replacement of Leu38 or Leu99 resulting in the loss of hydrophobic contacts to ubiquitin's Ile44- and Ile36-patch, respectively (Fig. 2i–k and Supplementary Fig. 2d, e).

Taken together, our MD simulations results suggest a model wherein ubiquitin binding to the USD promotes SPRTN activity by stabilizing an open conformation with an accessible active site.

### DNA- and ubiquitin-binding affect SPRTN's conformation synergistically

To experimentally test whether ubiquitin binds to the USD interface and whether ubiquitin binding affects SPRTN's interaction with DNA, we used NMR spectroscopy. Heteronuclear single quantum coherence (HSQC) spectra of SprT-BR showed well-dispersed peaks (Supplementary Fig. 3a, b). Comparisons with a ZBD-BR construct enabled us to transfer many chemical shifts based on our previous analysis of the ZBD-BR construct[21] (Supplementary Fig. 3b, c, see Figure legend for details). In particular, we could unambiguously assign Trp ε1 $^1$H,$^{15}$N resonances to the ZBD (Fig. 3, zoom-ins, orange labels) and protease domain (Fig. 3, zoom-ins, blue labels). Next, we compared NMR spectra of SprT-BR and SprT-BR-L99S, which superimposed very well (Supplementary Fig. 3d), except for those resonances in vicinity to the mutation site, indicating that structural integrity is not affected upon replacement of Leu99. Upon adding ubiquitin in five-fold excess, we observed some changes in the protease domain of SprT-BR spectra (Fig. 3a, blue boxes). In the L99S variant, the effects of ubiquitin addition were reduced, implying that they correspond to ubiquitin binding to SPRTN's USD interface (Fig. 3b, blue boxes). While the ubiquitin-induced effects were subtle and mostly affected resonances corresponding to the protease domain, we also observed line-broadening for signals corresponding to ZBD (Supplementary Fig. 3e, note Ile212). While Trp ε1 resonances were only marginally affected by the addition of ubiquitin (Fig. 3a, b, zoom-ins), the addition of an activating DNA structure in two-fold excess led to major spectral changes in ZBD-BR regions (Fig. 3c). DNA-induced line-broadening was comparable between WT and L99S constructs (Fig. 3d), demonstrating that alteration of the USD does not affect DNA binding. Strikingly, upon combined addition of both DNA and ubiquitin, severe line-broadening was observed in SprT-BR that was more pronounced than the individual effects of ubiquitin or DNA binding (Fig. 3e, red boxes), suggesting that the simultaneous binding of DNA and ubiquitin has synergistic effects on SPRTN's conformation. These effects were virtually absent in the L99S variant (Fig. 3f, red boxes). Consistently, addition of ubiquitin with a mutated Ile44-patch had little effect (Supplementary Fig. 4a, b).

Collectively, our NMR data indicate that ubiquitin amplifies the effects of DNA binding on SPRTN conformation allosterically by binding to the USD interface at the back of the protease domain. Interestingly, ubiquitin had only small effects on its own, implying that DNA binding occurs first and promotes ubiquitin binding at the USD.

### Ubiquitin stimulates DPC cleavage by binding to SPRTN's USD interface

To test whether DPC ubiquitylation stimulates SPRTN activity through binding to the USD interface, we produced full-length SPRTN with an L38S or L99S substitution. Both variants showed cleavage of unmodified HMCES$^{SRAP}$-DPCs to the same degree as the WT protein (Fig. 4a, compare lanes 3-5, with 6-8 (L38S) and 9-11 (L99S)). While DPC ubiquitylation increased overall activity also in USD mutant variants, the formation of smaller additional cleavage fragments (Cleaved DPC*) observed upon cleavage of ubiquitylated DPCs with the WT protease was reduced (L38S) or almost absent (L99S) (Fig. 4b, c, compare lanes 3-5 with lanes 6-8 (L38S) and lanes 9-11 (L99S)). Combination of the L38S and L99S substitution had no additional effects over the single L99S mutation (Supplementary Fig. 5a, b, compare lanes 6-8 (L99S) with lanes 9-11 (L38S + L99S)). These results suggest that DPC ubiquitylation promotes DPC cleavage through two distinct mechanisms. First, DPC ubiquitylation boosts overall cleavage by SPRTN independent of the USD interface (see Discussion). Second, DPC ubiquitylation allosterically activates SPRTN by binding to the USD interface, enabling the protease to cleave crosslinked proteins more efficiently.

### SUMO-targeted DPC ubiquitylation activates SPRTN in vitro and in cells

Encouraged by the strong effects observed using the synthetic DPC ubiquitylation system, we wanted to reconstitute SUMO-targeted DPC ubiquitylation using the enzymes that modify crosslinked proteins in cells. Therefore, we generated DPCs using full-length HMCES protein (HMCES$^{FL}$); we used HMCES$^{FL}$ because it contains a canonical SUMOylation site in its C-terminal tail that is absent in HMCES$^{SRAP}$ constructs. HMCES$^{FL}$-DPCs were incubated with the SUMOylation machinery, consisting of SUMO-E1, SUMO-E2, SUMO-E3 PIAS4, SUMO2 and ATP (Fig. 5a, b). Successful SUMOylation of the crosslinked protein was indicated by slower migrating HMCES$^{FL}$-DPC species that were absent in reactions lacking SUMO-E1 (Fig. 5b, compare lanes 3 and 4). For the subsequent ubiquitylation, SUMOylated DPCs were incubated with ubiquitin, ubiquitin-E1, ubiquitin-E2 UBE2D3 and the SUMO-targeted ubiquitin-E3 RNF4 (Fig. 5a, b). Ubiquitylation of SUMOylated DPCs was evident as further upshifts in gel migration and was confirmed by western blot (Fig. 5b, lane 7). We used MS to determine the identity of the ubiquitylated lysine residues and the involved ubiquitin linkages. We identified K48-, K63- and K11-linked ubiquitin chains on SUMOylated DPCs (Fig. 5c), as has been observed in cells[32]. Ubiquitin chains formed on various HMCES lysine residues and on three distinct SUMO2 lysine residues (Fig. 5d). Ubiquitylation was lost in the absence of ubiquitin-E1 or in the absence of SUMOylation (Fig. 5b, lanes 5 and 6 respectively), demonstrating bona fide SUMO-targeted DPC ubiquitylation.

Next, we incubated modified DPCs with SPRTN and FANCJ. Consistent with our results with the synthetic system, we observed enhanced cleavage of the ubiquitylated protein adduct by SPRTN, compared to unmodified DPCs and SUMOylated DPCs (Fig. 5e, compare lanes 3 and 5 with lane 7). Again, additional cleavage products appeared upon DPC ubiquitylation (Fig. 5e, Cleaved DPC*), which were reduced in variants with an altered USD interface (Fig. 5f, compare lanes 3-5 with lanes 6-8 (L38S) and lanes 9-11 (L99S)).

To test whether SUMO-targeted DPC ubiquitylation activates SPRTN also in cells, we monitored the cleavage of DNA methyltransferase 1 (DNMT1)-DPCs induced with 5-azadC[40]. DNMT1-DPCs are swiftly SUMOylated[41], triggering their ubiquitylation by RNF4[9,29,30] and TOPORS[31,32] and, subsequently, cleavage by SPRTN. While $SPRTN^{ΔC}$ cells are viable, they fail to efficiently cleave 5-azadC-induced DNMT1-DPCs[9]. Therefore, we complemented HeLa T-REx Flp-In cells expressing patient-mimicking $SPRTN^{ΔC}$ alleles from the endogenous locus with doxycycline-inducible full length SPRTN variants (WT, E112Q, L38S and L99S) and assessed cleavage of DNMT1-DPCs by the purification of x-linked proteins (PxP) assay (refs. 9,42, Fig. 5g and Methods). DNMT1-DPCs formed in all cell lines upon 5-azadC treatment (Fig. 5g). Following a 2-h chase in drug-free media, a specific cleavage band formed in $SPRTN^{ΔC}$ cells expressing SPRTN-WT but not in cells

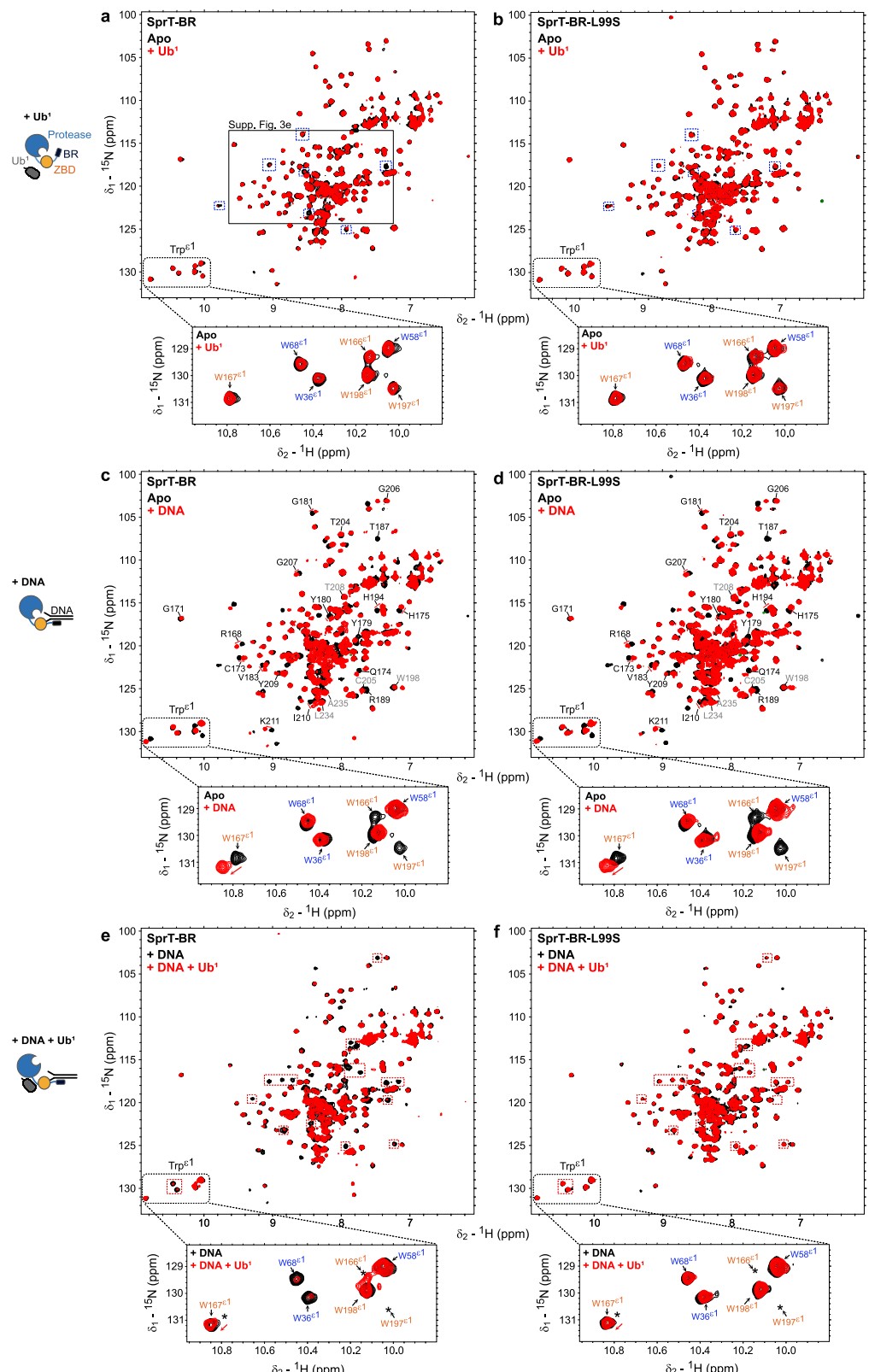

expressing catalytically inactive SPRTN-E112Q (Fig. 5g, red dots), as observed previously[9] (DPCs are still resolved in these cells because they are additionally targeted by proteasomal degradation[9,29]). SPRTN-dependent DNMT1-DPC cleavage was strongly reduced in cells expressing SPRTN-L38S or SPRTN-L99S (Fig. 5g, red dots), indicating that SUMO-targeted ubiquitylation promotes DPC cleavage in cells by activating SPRTN at the USD interface.

To corroborate this observation, we additionally assessed 5-azadC-induced SPRTN autocleavage (a marker of SPRTN activation) in the absence of DPC ubiquitylation. To abrogate ubiquitylation of DNMT1-DPCs, we depleted RNF4 using siRNA in HAP1 *TOPORS* knock-out cells. Simultaneous depletion of RNF4 and TOPORS resulted in a complete loss of SPRTN autocleavage (Supplementary Fig. 6a), confirming that DPC ubiquitylation is critical for efficient SPRTN activation in cells.

**Fig. 3 | DNA- and ubiquitin-binding affect SPRTN's conformation synergistically. a–f** Comparison of NMR spectra, highlighting Trp ε1 amide signals in $^1H,^{15}N$-HSQC experiments of SprT-BR and SprT-BR-L99S. Trp ε1 region is labeled and boxed (bottom). Resonance assignments corresponding to the Trp ε1's in the zinc-binding domain (ZBD) are shown in orange and those in the protease domain in blue. Broadened or shifted signals upon dsDNA addition are shown as asterisk. **a, b** SprT-BR (**a**) and SprT-BR-L99S (**b**) alone (= Apo) (black), with mono-ubiquitin (Ub[1]) (5x molar excess) (red). Minor changes are boxed in blue to highlight the

spectral differences between SprT-BR and SprT-BR-L99S upon adding Ub[1]. Zoom-in region in Supplementary Fig. 3e is marked with a black box (**b**). **c, d** SprT-BR (**c**) and SprT-BR-L99S (**d**) alone (black) (= Apo), with dsDNA (2x molar excess) (red). Some of the ZBD resonances affected by dsDNA are labeled in black while the unchanged are labeled in grey. **e, f** Superimpositions of SprT-BR (**e**) and SprT-BR-L99S (**f**) in the presence of dsDNA (2x molar excess) (black) and of both dsDNA (2x molar excess) and Ub[1] (5x molar excess) (red). Additional resonance changes upon adding Ub[1] to the dsDNA-bound SprT-BR are shown with red boxes.

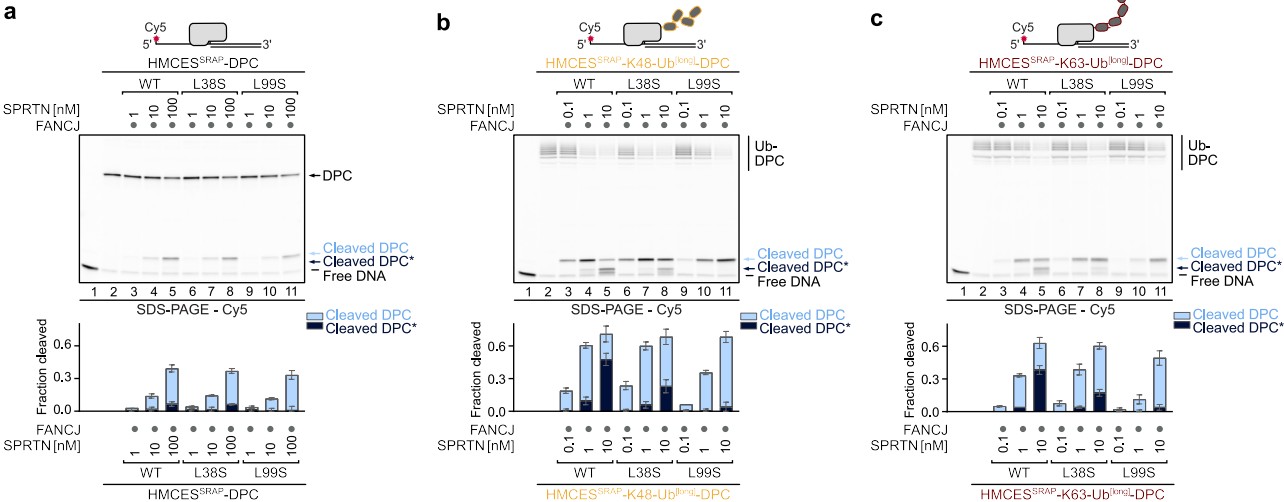

**Fig. 4 | The ubiquitin-dependent activation of SPRTN is mediated by the USD. a–c** Indicated HMCES[SRAP]-DPCs (10 nM) were incubated alone or in the presence of FANCJ (100 nM) and indicated concentrations (0.1–100 nM) and variants of SPRTN (WT, L38S, L99S) for 1 h at 30 °C. Quantification: bar graphs represent the mean ± SD of three independent experiments. Source data are provided as a Source Data file.

Given that DNMT1-DPC repair in cells is compromised upon replacement of critical USD residues and upon loss of SPRTN's *C*-terminal tail in RJALS SPRTN[ΔC] patient variants[9], we wanted to examine potential synergistic effects of both alterations using our reconstituted system. We compared cleavage of DPCs modified by SUMO-targeted ubiquitylation by SPRTN[FL] and SPRTN[ΔC] with intact or mutated USD interfaces. While SPRTN[ΔC] displayed only slightly reduced DPC cleavage compared to the WT enzyme (Fig. 5h, compare lanes 3-5 with lanes 9-11), the extent of cleavage by SPRTN[ΔC] was strongly reduced upon additional replacement of Leu99 by serine (Fig. 5h, compare lanes 9-11 and lanes 18-20). The synthetic cleavage defect of SPRTN[ΔC]-L99S was only partially explained by loss of the UBZ domain, given that SPRTN[ΔUBZ]-L99S variant displayed a less pronounced phenotype (Fig. 5h, lanes 15–17). Notably, the defect of SPRTN[ΔC] was specific to DPCs modified by SUMO-targeted ubiquitylation. DPCs modified using the synthetic ubiquitylation system were cleaved comparably well by SPRTN[ΔC] and the WT enzyme, while a USD mutant variant (L99S) displayed clear defects (Supplementary Fig. 6b and Discussion).

Taken together, our results suggest that SUMO-targeted DPC ubiquitylation allosterically activates SPRTN at the USD interface to promote DPC repair. Our in vitro data further imply that the ubiquitin-dependent activation of SPRTN is specifically important to support the residual cleavage of RJALS SPRTN[ΔC] patient variants towards DPCs modified by SUMO-targeted ubiquitylation.

### Ubiquitin-dependent activation of SPRTN maintains genome stability in Ruijs-Aalfs syndrome

Next, we wanted to determine whether the ubiquitin-dependent activation of SPRTN at the USD interface is important to maintain the residual function of SPRTN[ΔC] patient variants in cells. To this end, we

complemented conditional *Sprtn[F/-] CreER[T2]* knock-out mouse embryonic fibroblasts (MEFs) with either an empty vector (EV) or different human SPRTN variants (FL and ΔC) tagged with a *C*-terminal Strep-tag (Supplementary Fig. 7a, b). Of note, SPRTN[ΔC] variants expressed at much higher levels than the WT enzyme (Supplementary Fig. 7a, b), as previously observed in RJALS patients[27]. Loss of endogenous *Sprtn* was induced by 4-hydroxytamoxifen (4-OHT), with the solvent MeOH serving as control (Supplementary Fig. 7c, d), and resulted in diverse phenotypes including growth arrest (Fig. 6a, b), formation of micronuclei and chromatin bridges (Fig. 6c–e), as wells as arrest in the G2/M phase of the cell cycle (Supplementary Fig. 7e–h), as described previously[17]. All phenotypes were rescued by expression of human WT SPRTN but not by catalytically inactive SPRTN-E112Q (Fig. 6a and d). Also, expression of SPRTN[ΔC] complemented all phenotypes induced by *Sprtn* knock-out (Fig. 6b and e). While the replacement of USD residues Leu38 or Leu99 had no effect on the ability of full-length SPRTN to complement cell fitness and cell cycle defects upon loss of mouse *Sprtn* (Fig. 6a and Supplementary Fig. 7e), loss of Leu99 resulted in intermediate growth defects and G2/M arrest in SPRTN[ΔC] (Fig. 6b and Supplementary Fig. 7f). These defects were accompanied by severe signs of genome instability, observed as micronuclei and chromatin bridges in cells expressing SPRTN[ΔC]-L99S (Fig. 6c and e).

Collectively, these experiments demonstrate that SPRTN's USD interface, and thus the allosteric activation of SPRTN by ubiquitin, is critical to maintain fitness and genome stability in cells expressing truncated RJALS patient variants.

## Discussion

Over the last decade, DPC repair has emerged as a conserved cellular process that is essential for maintaining genome stability[7]. Since the

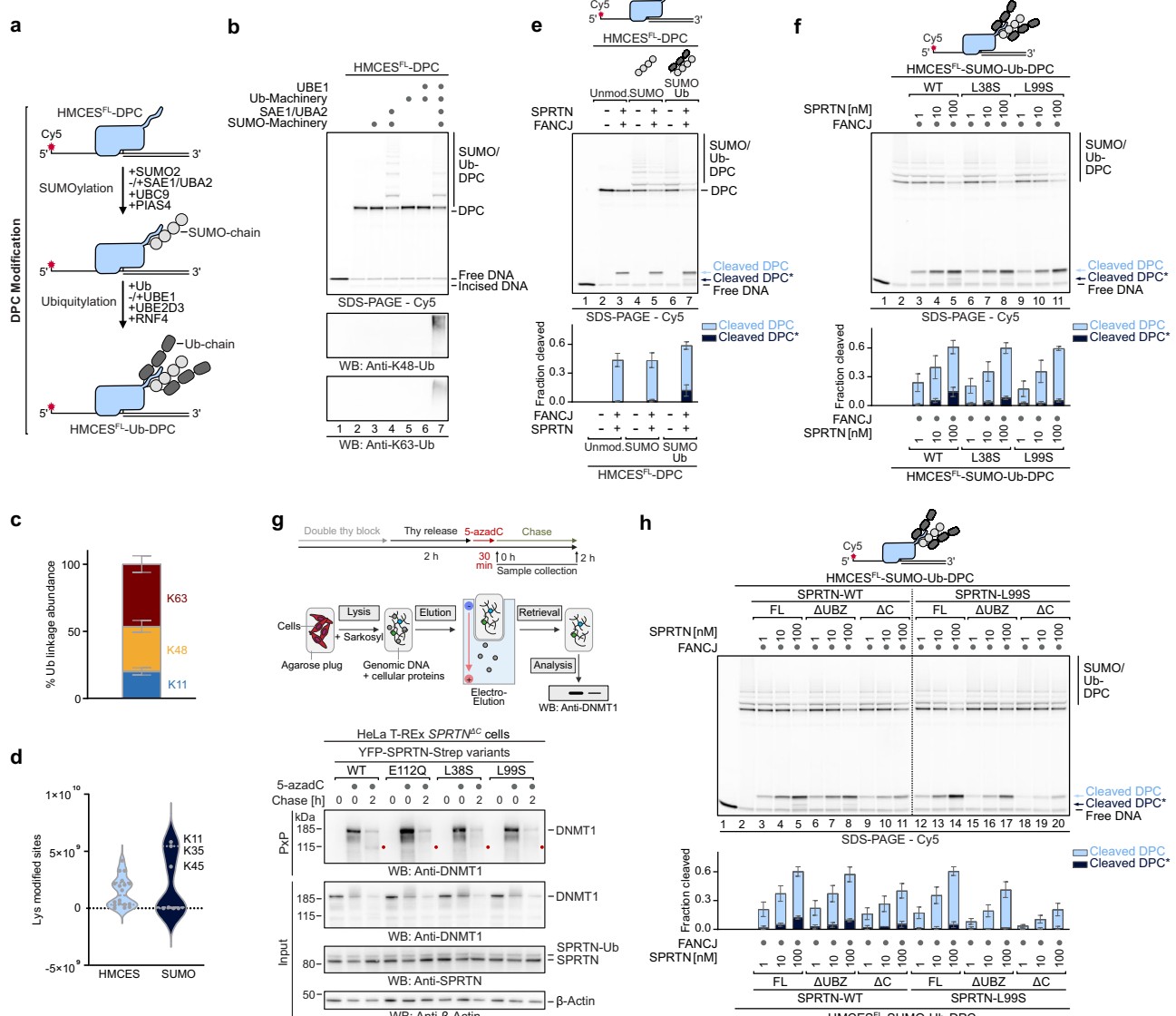

**Fig. 5 | SUMO-targeted DPC ubiquitylation activates SPRTN. a** Schematic of SUMO-targeted ubiquitylation of HMCES[FL]-DPCs used in b-f and h. HMCES[FL]-DPCs were incubated alone or in the presence of SUMO2, UBC9 and PIAS4, with or without SAE1/UBA2 for 30 min at 37 °C. Next unmodified or SUMOylated HMCES[FL]-DPCs were incubated alone or in the presence of ubiquitin (Ub), RNF4, UBE2D3, with or without UBE1 for 30 min at 37 °C. **b** SUMO-targeted ubiquitylated HMCES[FL]-DPCs generated as described in (**a**), separated by denaturing SDS-PAGE and immunoblotting. Source data are provided as a Source Data file. **c** Mass spectrometry analysis of ubiquitin linkages formed by SUMO-targeted ubiquitylation of HMCES[FL]-DPCs. Bar chart shows the mean ± SD of four biological replicates. **d** Mass spectrometry analysis of lysine residues within HMCES or SUMO modified upon SUMO-targeted ubiquitylation. Violin blots show the mean ± SD of four biological replicates. **e** Indicated HMCES[FL]-DPCs (10 nM) were incubated alone or in the presence of FANCJ (100 nM) and SPRTN (100 nM) for 1 h at 30 °C. Quantifications: bar graphs represent the mean ± SD of three independent experiments. Source data are provided as a Source Data file. **f** Indicated HMCES[FL]-DPCs (10 nM) were incubated

alone or in the presence of FANCJ (100 nM) and indicated concentrations (1-100 nM) and variants of SPRTN (WT, L38S, L99S) for 1 h at 30 °C. Quantifications: bar graphs represent the mean ± SD of three independent experiments. All samples derive from the same experiment and gels were processed in parallel. Source data are provided as a Source Data file. **g** HeLa-TREx *SPRTN[ΔC]* Flp-In cells complemented with indicated YFP-SPRTN[FL]-Strep-tag variants were treated as depicted (top) with 5-azadC (10 μM) and harvested at indicated time points. DNMT1-DPCs were isolated using PxP (middle, see Methods) and analyzed by immunoblotting (bottom). Shown is a representative of three independent experiments. Source data are provided as a Source Data file. **h** Indicated HMCES[FL]-DPCs (10 nM) were incubated alone or in the presence of FANCJ (100 nM) and indicated concentrations (1–100 nM) and variants of SPRTN (FL-WT/L99S, ΔUBZ-WT/L99S, ΔC-WT/L99S) for 1 h at 30 °C. Quantifications: bar graphs represent the mean ± SD of three independent experiments. All samples derive from the same experiment and gels were processed in parallel. Source data are provided as a Source Data file.

identification of dedicated DPC proteases in yeast and humans[10,20,43–48], it has remained enigmatic how specificity for crosslinked protein adducts is achieved. The DPC protease SPRTN features a bipartite DNA binding module, consisting of ZBD and BR, which provides a first layer of specificity by restricting activity to the cleavage of DPCs near ssDNA-dsDNA junctions and other structures with single- and double-stranded features[14,19,21]. However, because such structures occur

frequently across the genome, SPRTN's DNA structure-specific activity alone is insufficient to explain how the protease achieves specificity.

Our study reveals that SPRTN activation is controlled by the ubiquitylation of the crosslinked protein by ubiquitin-E3 ligases. By reconstituting DPC ubiquitylation in vitro, we observed that this modification stimulates SPRTN activity by up to two orders of magnitude, regardless of ubiquitin chain linkage type. Our results indicate

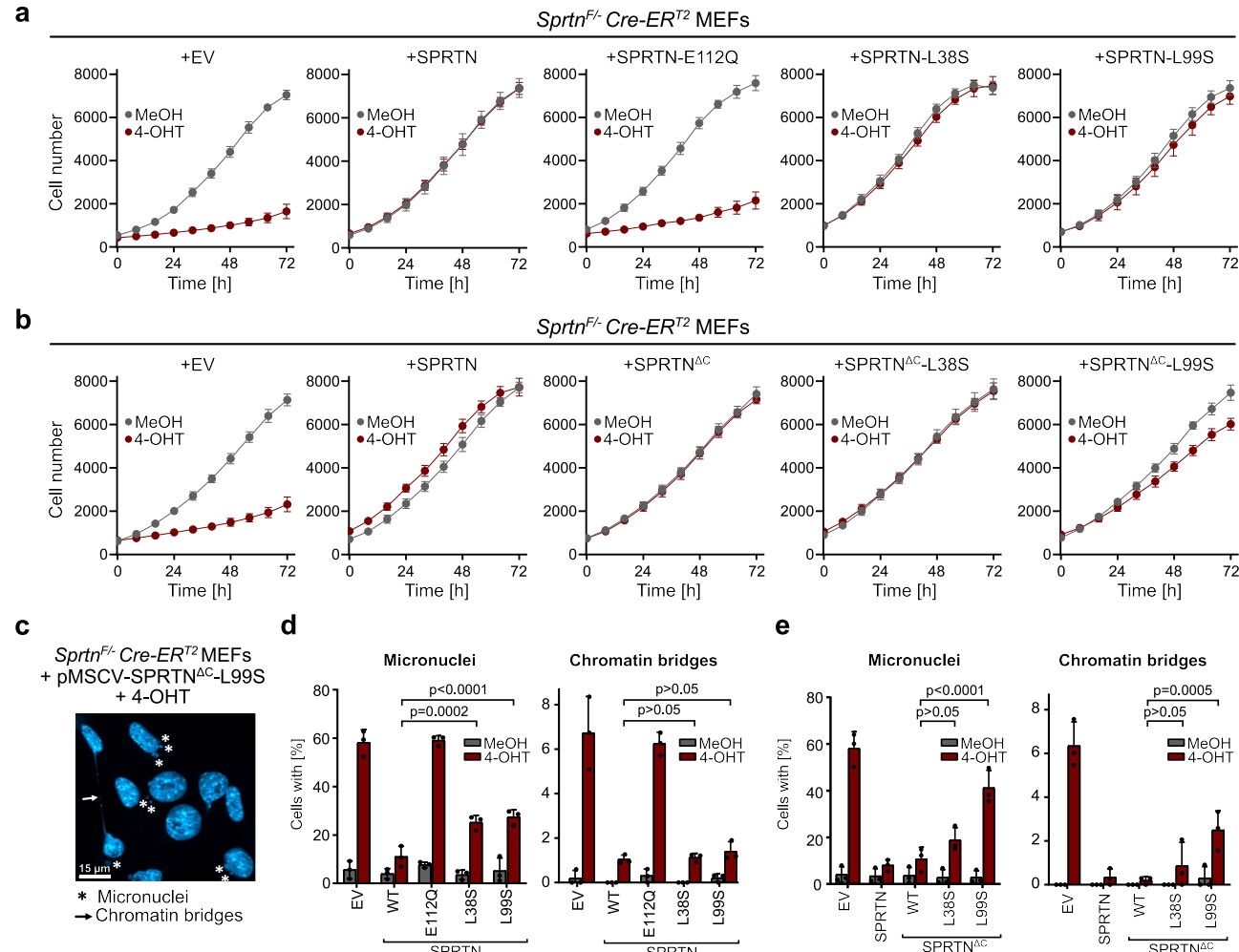

**Fig. 6 | Ubiquitin-dependent activation of SPRTN maintains genome stability in Ruijs-Aalfs syndrome. a**, **b** Proliferation of *Sprtn*[F/-] Cre-ER[T2] mouse embryonic fibroblasts (MEFs) complemented with indicated SPRTN variants or empty vector (EV, pMSCV) treated with methanol (MeOH) or (Z)−4-hydroxytamoxifen (4-OHT) (2 μM) for 48 h. After seeding, cell numbers were counted at indicated time points. Values are the mean ± SD of eight technical replicates. Shown is a representative of three independent experiments. Source data are provided as a Source Data file. **c** Image showing micronuclei (asteriks) and chromatin bridges (arrow) in *Sprtn*[F/-] Cre-ER[T2] MEFs + pMSCV-SPRTN[ΔC]-L99S treated with 4-OHT (2 μM) for 48 h. DNA was visualized by DAPI staining. Scale bar corresponds to 15 μm. **d**, **e** Quantification of micronuclei and chromatin bridges formation in *Sprtn*[F/-] Cre-ER[T2] MEFs

complemented with indicated SPRTN variants or EV (pMSCV) treated with MeOH or 4-OHT (2 μM) for 48 h. DNA was visualized by DAPI staining. Bar graphs show the mean ± SD of three independent experiments. The p values were calculated using a two-way ANOVA with Dunnett's multiple comparison test. P values: **d** Micronuclei (left): SPRTN-WT vs. SPRTN-L38S = 0.0002; SPRTN-WT vs. SPRTN-L99S < 0.0001. Chromatin bridges (right): SPRTN-WT vs. SPRTN-L38S = 0.9992; SPRTN-WT vs. SPRTN-L99S = 0.8634. **e** Micronuclei (left): SPRTN[ΔC]-WT vs. SPRTN[ΔC]-L38S = 0.1411; SPRTN[ΔC]-WT vs. SPRTN[ΔC]-L99S < 0.0001. Chromatin bridges (right): SPRTN[ΔC]-WT vs. SPRTN[ΔC]-L38S = 0.4745; SPRTN[ΔC]-WT vs. SPRTN[ΔC]-L99S = 0.0005. Source data are provided as a Source Data file.

two distinct mechanisms for the activation of SPRTN. On the one hand, ubiquitin activates SPRTN by binding to the USD interface at the back of the protease domain. As a result, the enzyme processes DPCs to a much greater extent, which may be crucial for enabling TLS polymerases to efficiently bypass the remaining peptide adduct during replication-coupled DPC repair[10]. Our MD simulations and NMR data further suggest that the USD-ubiquitin interaction stabilizes a DNA-binding induced open conformation of the enzyme, exposing its active site. In addition, DPC ubiquitylation stimulates overall DPC cleavage independent of the USD interface. The underlying mechanisms remain unclear and are an exciting topic for future research.

These insights help explain why cells ubiquitylate DPCs[9,10,12–15,28–32] and why ubiquitylated DPCs accumulate in cells following SPRTN depletion[33]. Of note, the observation that SPRTN-dependent cleavage can occur without DPC ubiquitylation in frog egg extracts[14,15] is not necessarily inconsistent with our findings. It is plausible that the DPC cleavage observed in egg extract in the absence of ubiquitylation

originates from SPRTN's basal, ubiquitin-independent activity, which is also evident in our assays.

Consistently, while amino acid substitutions within the USD interface substantially reduced cleavage of ubiquitylated DPCs in vitro and of DNMT1-DPCs in cells, they did not completely abolish SPRTN function. SPRTN with a replacement of the USD residue Leu99, which consistently showed stronger effects compared to replacing Leu38, suppressed almost all phenotypes caused by the loss of *Sprtn* in MEFs. The same is true for the RJALS SPRTN[ΔC] patient variant. Thus, only a minimal amount of SPRTN activity appears to be necessary to fulfil its essential role in suppressing genome instability. The critical role of the USD became evident when Leu99 was replaced in SPRTN[ΔC], resulting in cell fitness defects and formation of micronuclei and chromatin bridges in mitosis.

The synthetic effect observed between the combined loss of SPRTN's *C*-terminal tail and a functional USD interface, is only partially explained by the loss of the UBZ domain. While the UBZ is required for

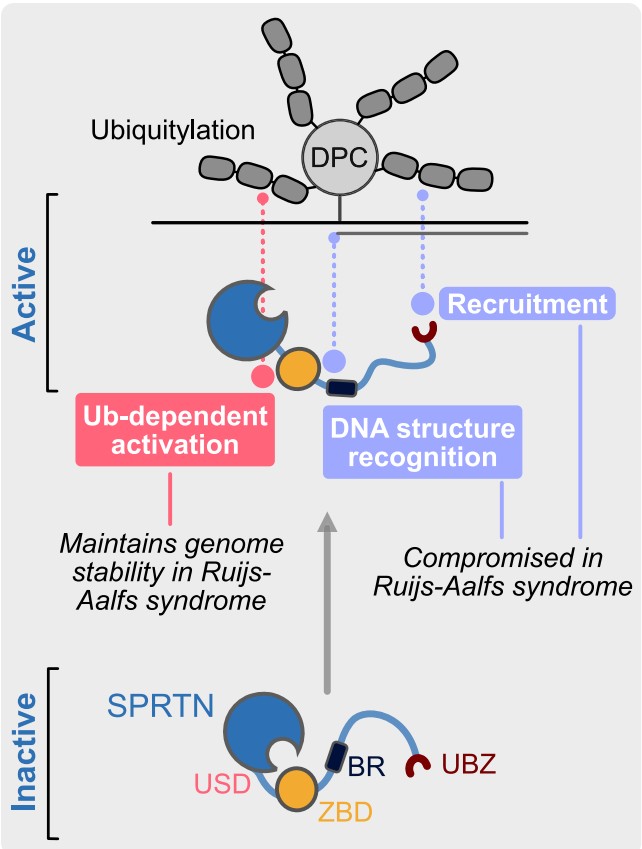

**Fig. 7 | 'Triple-lock' model for SPRTN activation.** The ubiquitin-binding zinc finger (UBZ) recruits SPRTN to ubiquitylated DPCs. Binding of both DNA-binding domains, zinc-binding domain (ZBD) and the basic region (BR) to activating DNA structures induces an open conformation. This open conformation is stabilized by ubiquitin binding to the ubiquitin binding interface at the SprT domain (USD). Recruitment and DNA structure recognition are compromised in Ruijs-Aalfs syndrome patients, which therefore fully rely on the Ub-dependent activation via the USD to maintain genome stability.

efficient DPC cleavage in cells[9] and egg extracts[14], it showed weaker defects in processing ubiquitylated DPCs in combination with the L99S substitution than the corresponding SPRTN^ΔC-L99S variant. In addition to lacking the UBZ domain, SPRTN^ΔC also exhibits reduced DNA binding compared to the FL protein[20,44], which may contribute to its reliance on the USD for full functionality. Based on these considerations, we propose a partially speculative 'triple-lock' model in which SPRTN activity is controlled by at least three mechanisms (Fig. 7). First, the UBZ supports SPRTN function by recruiting it to ubiquitylated DPCs, as previously suggested[9,14]. This recruitment function is likely more important in the crowded environment of a cell than in our in vitro experiments, explaining why the loss of the UBZ had no or only weak phenotypes in our assays. Second, the binding of an activating DNA structure induces an open conformation of SPRTN. Third, this open, active conformation is further stabilized by binding of ubiquitylated DPCs to SPRTN's USD interface, facilitating rapid and complete proteolysis of the crosslinked protein adduct.

This model offers a potential explanation for why SPRTN^ΔC displayed defects in processing DPCs modified by SUMO-targeted ubiquitylation but not of DPCs modified using the synthetic ubiquitylation system. In the synthetic system, the DPC is modified exclusively at the C-terminal ubiquitin tag[35]. In contrast, our MS analysis revealed that SUMO-targeted ubiquitylation affects multiple lysine residues within the DPC and the SUMO chain. Some of these ubiquitylation events may

hinder SPRTN function by interfering with efficient DNA binding. Thus, SPRTN's full DNA binding capacity is likely required in this context. In cells, potential steric hindrance of DNA access by SPRTN due to DPC ubiquitylation may be overcome by p97-dependent unfolding of the crosslinked protein[49].

The ability of SPRTN to be activated by both K48- and K63-linked ubiquitin chains raises an hypothesis as to why SPRTN is essential, despite acting redundantly with the proteasome in most experimental systems investigated so far[9,14,15]. Given that the proteasome mainly targets substrates modified with K48-linked ubiquitin[50], key endogenous substrates of SPRTN may be modified by K63-linked ubiquitin and are consequently not amenable to proteasomal degradation.

In conclusion, we have demonstrated that ubiquitin does not simply recruit SPRTN but allosterically activates the enzyme, which is essential for maintaining genome stability in RJALS patients. This sophisticated mechanism likely evolved to constrain the potentially toxic activity of SPRTN and represents a unique ubiquitin-dependent regulatory principle in DNA repair.

## Methods

### Mammalian cell lines

HeLa TREx Flp-In *SPRTN^ΔC* cells (ref. 9) stably expressing YFP-SPRTN-Strep-tag variants were generated using the Flp-In system (pOG44; V600520, Thermo Scientific) according to manufacturer's instructions and selected in Hygromycin B (150 μg/mL) (10687010, Thermo Scientific). Protein expression was induced by overnight incubation with doxycycline hyclate (DOX) (D9891, Sigma) (final concentration 1 μg/mL). HeLa TREx Flp-In *SPRTN^ΔC* cells were grown in Dulbecco's Modified Eagle Medium (DMEM) supplemented with 10% (v/v) fetal bovine serum (FBS).

HAP1 wild-type (WT) cells (C631, Horizon Discovery) and HAP1 *TOPORS* knock-out (KO) cells (HZGHC008005c006, Horizon Discovery) were grown in Iscove's Modified Dulbecco's Medium (IMDM) supplemented with 10% (v/v) FBS and 1% (v/v) Penicillin-Streptomycin-Glutamine (PSG).

*Sprtn^F/-* mouse embryonic fibroblasts (MEFs) (H7) immortalized with SV40 large T and transduced with a retroviral vector expressing Cre-ER^T2 (ref. 17) were cultured in DMEM supplemented with 10% (v/v) FBS. *Sprtn* KO was induced by treating 4×10^5 cells with methanol (MeOH) (vehicle control) or 2 μM (Z)−4-hydroxytamoxifen (4-OHT) (H-t7904, Sigma) for 48 h. Conversion of the floxed *Sprtn* allele **f** to the KO allele (-) was verified by PCR using WT- (5′-GTGCTGGGATCTGCAC CTAT-3′) and KO-specific (5′-CCATCAGGGACGTTTTCTTG-3′) forward primers and a common reverse primer (5′-TGCACAGCTGTAAACCC TTG-3′). PCR conditions were 35 cycles of 94 °C for 30 s, 60 °C for 30 s, and 72 °C for 1 min. PCR products are 527 bp and 278 bp for the floxed and the KO alleles, respectively. For exogenous expression of human SPRTN in MEFs, cells were infected with retroviral vectors produced in HEK293T/17 (CRL-11268, ATCC) by co-transfecting pMSCV.hyg-SPRTN-Strep with gag-pol and VSV-G packaging plasmids. Infected cells were selected with Hygromycin B (200 μg/mL) (10687010, Thermo Scientific) for 8 days.

To confirm protein expression, cells were lysed in NP-40 lysis buffer (50 mM Tris-HCl, pH 7.4, 150 mM NaCl, 0.1% NP-40 (IGEPAL) (I8896, Sigma), 10% glycerol, 5 mM EDTA (BP118-500, Fisher BioReagents), 50 mM NaF, 1 mM Na₃VO₄) supplemented with protease inhibitor cocktail (P8849, Sigma). Cell lysates containing 30 μg protein were resolved on SDS-PAGE gels (4-12% Bis-Tris NuPAGE, Thermo Scientific) using MOPS buffer. Resolved proteins were subsequently immunoblotted using anti-SPRTN antibody (1:500) (6F2) (ref. 26) and anti-β-actin antibody (1:1000) (Sc47778, Santa Cruz Biotechnology).

### Protein expression and purification

**SPRTN.** Amino acid replacements and deletions for SPRTN variants were generated using the Q5-site-directed mutagenesis kit (E0554S,

NEB). Recombinant SPRTN (Full-length and ΔUBZ – WT or in combination with L38S, L99S amino acid replacements) protein was expressed in BL21 (DE3) *E. coli* cells (C600003, Thermo Scientific) and purified as previously described with slight modifications[21].

BL21 (DE3) *E. coli* cells were grown at 37 °C in Terrific broth (TB) medium (prepared with tap water) until they reached OD 0.7. Protein expression was induced by addition of 0.5 mM Isopropyl-β-D-thiogalactoside (IPTG) (I6758, Sigma) overnight at 18 °C. The next day, cells were harvested, snap-frozen in liquid nitrogen and stored at −80 °C. All subsequent steps were carried out at 4 °C. For protein purification, cell pellets were resuspended in buffer A (50 mM HEPES/KOH pH 7.2, 500 mM KCl, 1 mM MgCl$_2$, 10% Glycerol, 0.1% IGEPAL (I8896, Sigma), 0.04 mg/mL Pefabloc SC (76307, Sigma), cOmplete EDTA-free protease inhibitor cocktail tablets (4693132001, Roche), 1 mM Tris(2-carboxyethyl)phosphine hydrochloride (TCEP)) (HN95.3, Roth) and lysed by sonication. Cell lysate was incubated with smDNAse (45 U/mL lysate) (MPI for Biochemistry) for 30 min on a roller prior to removal of cell debris by centrifugation at 18,000 × *g* for 30 min. Cleared supernatant was filtered using syringe filters (PVDF, 0.22 μm) and applied to Strep-Tactin®XT 4Flow® high-capacity cartridges (2-5028-001, IBA Lifesciences), washed with 3 column volumes (CV) of buffer A and 4 CV of buffer B (50 mM HEPES/KOH pH 7.2, 500 mM KCl, 10% Glycerol, 1 mM TCEP (HN95.3, Roth)). Proteins were eluted in 6 CV buffer C (50 mM HEPES/KOH pH 7.2, 500 mM KCl, 10% Glycerol, 1 mM TCEP (HN95.3, Roth) and 50 mM Biotin). Eluted proteins were further applied to HiTrap Heparin HP affinity columns (17040701, Cytiva) and washed with 3 CV buffer B before eluting in buffer D (50 mM HEPES/KOH pH 7.2, 1 M KCl, 10% Glycerol, 1 mM TCEP (HN95.3, Roth)). Eluted fractions containing recombinant SPRTN protein were desalted against buffer B using PD-10 desalting columns (17085101, Cytiva). The affinity tag was cleaved off at 4 °C overnight by addition of His-tagged TEV protease (ref. [37]) with 1:10 mass ratio. Cleaved recombinant SPRTN protein was further purified by size exclusion chromatography (SEC) using a HiLoad 16/600 Superdex 200 pg column (28989335, Cytiva) equilibrated in buffer B (50 mM HEPES/KOH pH 7.2, 500 mM KCl, 10% Glycerol, 1 mM TCEP (HN95.3, Roth)). Eluted proteins were concentrated with 10 kDa cut-off Amicon Ultra centrifugal filters (UFC801096, Merck) before aliquoting, snap-freezing in liquid nitrogen and storage at −80 °C.

Following SPRTN purification, metalation of the protein was examined by Inductively Coupled Plasma Optical Emission Spectrometry (ICP-OES) (see Supplementary Table 1), which confirmed correct metalation with three Zn$^{2+}$ ions per full-length SPRTN molecule.

For truncated SPRTN variants smaller than 30 kDa including SPRTN$^{\Delta C}$ (WT or L99S), SprT-BR (WT, L99S, W36G and W58G), ZBD-BR and ZBD, a Strep-tagged TEV protease (ref. [9]) was used. Prior to SEC, Strep-tagged TEV protease, residual uncleaved protein and the cleaved Tag were removed by a Strep-Tactin®XT 4Flow® high capacity cartridges (2-5028-001, IBA Lifesciences)[9].

For NMR experiments, SprT-BR (L99S, W36G and W58G), ZBD, and ZBD-BR were expressed in $^{15}$N- or $^{13}$C-/$^{15}$N-containing media. Here, cells were grown to OD 0.4, before the temperature of the incubator was lowered to 18 °C and MnCl$_2$ was added to a final concentration of 1.5 mM. Once OD 0.7 was reached protein expression was induced with 0.5 mM IPTG (I6758, Sigma) and performed overnight at 18 °C. For SEC, buffer E (50 mM HEPES/KOH pH 7.2, 500 mM KCl, 1% Glycerol, 2 mM TCEP (HN95.3, Roth), pH 7.2) was used.

**Mono-Ubiquitin.** For purification of mono-ubiquitin (Ub$^1$) a plasmid encoding Ub$^1$ with a *N*-terminal His6-Tag was provided by Brenda Schulman (MPI for Biochemistry, Martinsried, Germany). Ub$^1$-I44A was generated by introducing point mutations using the Q5-site-directed mutagenesis kit (E0554S, NEB). Protein was expressed in Rosetta *E. Coli* cells (70-954-3, Sigma), grown at 37 °C in TB (prepared with tap water) to OD 0.7. Protein expression was induced with 0.5 mM IPTG (I6758,

Sigma) overnight at 18 °C. Cells were harvested the next day and directly resuspended in buffer A (50 mM Tris-HCl pH 7.5, 250 mM NaCl) (20 mL/ L culture), snap-frozen in liquid nitrogen and stored at −80 °C. All subsequent steps were carried out at 4 °C. For protein purification, cell lysates were thawed and Pefabloc SC (0.04 mg/mL) (76307, Sigma), MgCl$_2$ (1 mM) and smDNAse (45 U/mL lysate) (MPI for Biochemistry) were added. Cells were lysed by sonication and incubated for 30 min on a roller prior to removal of cell debris by centrifugation at 50,000 × *g* for 30 min. Clarified lysate was filtered using syringe filters (PVDF, 0.22 μm) and mixed with Ni-NTA Agarose (30250, Qiagen) equilibrated in buffer A and incubated for 1 h on a roller to allow binding. The beads were transferred to a gravity flow column, washed with 15 CV of buffer A and protein was eluted in fractions of 1 CV each with buffer B (50 mM Tris-HCl pH 7.5, 250 mM NaCl, 300 mM imidazole (3899.1, Roth)). Fractions were checked via SDS-PAGE and Coomassie-based staining for presence of Ub$^1$. Ub$^1$-containing fractions were pooled and after addition of GST-tagged 3C-protease (0.5 mg/L culture) (MPI for Biochemistry), dialyzed against buffer A overnight. Cleaved protein was passed through Ni-NTA Agarose (30250, Qiagen) the next day for removal of uncleaved protein and His6-tag. The flow-through was collected, concentrated to 1 mL and loaded on a Superdex 200 Increase 10/300 GL column (28990944, Cytiva) equilibrated in buffer C (50 mM HEPES/KOH pH 7.2, 500 mM KCl, 1% Glycerol, 2 mM TCEP (HN95.3, Roth)). Eluted protein was concentrated with 10 kDa cut-off Amicon ultra centrifugal filters (UFC801096, Merck) before aliquoting, snap-freezing in liquid nitrogen and storage at −80 °C.

**FANCJ.** Recombinant FANCJ protein was expressed in High Five™ cells (B85502, Thermo Scientific) and purified as previously described[38].

**HMCES$^{SRAP}$.** Recombinant HMCES$^{SRAP}$, protein was expressed in BL21 (DE3) *E. coli* cells (C600003, Thermo Scientific) and purified as previously described[38], using TB (prepared with tap water). For synthetic ubiquitylation of HMCES$^{SRAP}$, a sequence encoding for Ub$^1$(G76V) followed by an FKBP-domain, including linkers and a 3C-protease cleavage site was codon optimized for bacterial expression and inserted at the *C*-terminal end of HMCES$^{SRAP}$, in front of the His6-tag, in the pNIC_HMCES$^{SRAP}$ plasmid. Purification followed protocols described for HMCES$^{SRAP}$ and the final protein was further processed as described below.

**HMCES$^{FL}$.** Recombinant HMCES$^{FL}$ protein was expressed in BL21 (DE3) *E. coli* cells (C600003, Thermo Scientific) and purified as previously described[38], analogously to recombinant SPRTN using TB (prepared with tap water).

**UBC9.** For purification of recombinant UBC9, the open reading frame was codon optimized and cloned in a pBAD plasmid carrying a *N*-terminal His6-tag. Protein was expressed in BL21(DE3) *E. coli* cells (C600003, Thermo Scientific) and grown in TB (prepared with tap water) at 37 °C to OD 0.7 before induction of protein expression with 0.1% L-arabinose (A3256, Sigma) at 18 °C overnight. Cells were harvested the next day, snap-frozen in liquid nitrogen and stored at −80 °C. All subsequent steps were performed at 4 °C. For protein purification, cell pellets were thawed, resuspended in buffer A (20 mM HEPES/KOH pH 7.0, 2 mM Mg(OAc)$_2$, 300 mM KOAc, 10% glycerol, 30 mM imidazole (3899.1, Roth), 0.1% IGEPAL (I8896, Sigma), 1 mM TCEP (HN95.3, Roth), cOmplete protease inhibitor (4693132001, Roche), 0.04 mg/mL Pefabloc SC (76307, Sigma), 1 mg/mL lysozyme (8259.3, Roth), 45 U/mL smDNAse (MPI for Biochemistry) and incubated on a roller for 30 min. The lysate was sonicated for 15 min prior to cell debris removal by centrifugation at 18,000 x *g* for 40 min. Clarified lysate was filtered using syringe filters (PVDF, 0.22 μm) and incubated with Ni-NTA agarose (30250, Qiagen) on a roller for 1 h at

4 °C. The beads were transferred to a gravity flow column and washed with 15 CV buffer B (20 mM HEPES/KOH pH 7.0, 2 mM Mg(OAc)$_2$, 300 mM KOAc, 10% glycerol, 30 mM imidazole (3899.1, Roth)) before elution in 2 CV buffer C (20 mM HEPES/KOH pH 7.0, 2 mM Mg(OAc)$_2$, 300 mM KOAc, 10% glycerol, 300 mM imidazole (3899.1, Roth)). The His6-tag was cleaved by the addition of His-tagged TEV protease (1 mg/L culture) (ref. 37) and dialyzed overnight against buffer D (20 mM HEPES/KOH pH 7.0, 2 mM Mg(OAc)$_2$, 300 mM KOAc). The next day, cleaved protein was passed through Ni-NTA agarose (30250, Qiagen) to remove His-tagged TEV protease, residual uncleaved protein and His6-Tag. Flow-through was concentrated to 1 mL and loaded on a Superdex 200 Increase 10/300 GL column (28990944, Cytiva) equilibrated in buffer E (20 mM HEPES/KOH pH 7.0, 2 mM Mg(OAc)$_2$, 300 mM KOAc, 10% glycerol). Eluted protein was concentrated with 10 kDa cut-off Amicon ultra centrifugal filters (UFC801096, Merck) before aliquoting, snap-freezing in liquid nitrogen, and storage at −80 °C.

**PIAS4**. The open reading frame of PIAS4 was codon optimized and cloned in a pNIC plasmid in frame with a *N*-terminal TwinStrep-ZB-tag. Recombinant PIAS4 protein was expressed in BL21 (DE3) *E. coli* cells (C600003, Thermo Scientific), grown in TB (prepared with tap water) at 37 °C to OD 0.7 before induction with 1 mM IPTG (I6758, Sigma) and expression at 18 °C overnight. Protein purification was done analogously to SPRTN.

**UBE2D3**. For purification of UBE2D3, the open reading frame was codon optimized and cloned into a pDEST17 plasmid carrying an *N*-terminal His6-tag. UBE2D3 was expressed in BL21(DE3) *E. coli* cells (C600003, Thermo Scientific), grown in TB media (prepared with tap water) to an OD of 0.7 at 37 °C. Expression was induced by the addition of 0.5 mM IPTG (I6758, Sigma) for 3 h at 37 °C. Cells were harvested, snap frozen in liquid nitrogen and stored at −80 °C. All subsequent steps were performed at 4 °C. For protein purification, cell pellets were thawed and resuspended in 50 mL buffer A (50 mM Na$_2$HPO$_4$/NaH$_2$PO$_4$ pH 8.0, 150 mM NaCl, 10 mM imidazole (3899.1, Roth), 1 mM TCEP (HN95.3, Roth), cOmplete protease inhibitor (4693132001, Roche), 0.04 mg/mL Pefabloc SC (76307, Sigma)) and lysed by sonication. DNA was digested by the addition of smDNAse (45 U/mL lysate) (MPI for Biochemistry) for 30 min on a roller, followed by centrifugation at 18,000 x *g* for 30 min to remove cell debris. Clarified lysate was filtered using syringe filters (PVDF, 0.22 µm) and incubated with Ni-NTA agarose (30250, Qiagen) on a roller for 1 h at 4 °C. The beads were washed with 20 mL buffer B (50 mM Na$_2$HPO$_4$/NaH$_2$PO$_4$ pH 8.0, 500 mM NaCl, 20 mM imidazole (3899.1, Roth), 1 mM TCEP (HN95.3, Roth)) and eluted in 5 mL buffer C (50 mM Na$_2$HPO$_4$/NaH$_2$PO$_4$ pH 8.0, 500 mM NaCl, 250 mM imidazole, 1 mM TCEP (HN95.3, Roth)). The eluted protein was dialyzed against buffer D (20 mM Tris/HCl pH 7.5, 150 mM NaCl, 10% glycerol, 0.5 mM TCEP (HN95.3, Roth)) overnight followed by SEC on a Superdex 200 Increase 10/300 GL column (28990944, Cytiva) equilibrated in buffer D. Eluted protein was concentrated using 10 kDa cut-off Amicon ultra centrifugal filters (UFC801096, Merck) before aliquoting, snap-freezing in liquid nitrogen and storage at −80 °C.

**RNF4**. For purification of recombinant RNF4, the open reading frame was codon optimized and cloned in a pNIC plasmid in frame with a *N*-terminal TwinStrep-ZB-tag. RNF4 protein was expressed in BL21 (DE3) *E. coli* cells (C600003, Thermo Scientific), grown in TB (prepared with tap water) and purified analogously to SPRTN. For SEC, buffer E (50 mM HEPES/KOH pH 7.2, 150 mM KCl, 10% glycerol, 1 mM TCEP (HN95.3, Roth)) was used.

**SUMO2**. For purification of recombinant SUMO2, the open reading frame was codon optimized and cloned in a pBAD plasmid carrying a

*N*-terminal His6-tag. SUMO2 was expressed in BL21 (DE3) *E. coli* cells (HN95.3, Roth) and grown in TB (prepared with tap water) at 37 °C to OD 0.7 before induction with 0.02% L-arabinose (A3256, Sigma) and expression at 18 °C overnight. Cells were harvested the next day, snap-frozen in liquid nitrogen and stored at −80 °C. All subsequent steps were performed at 4 °C. For protein purification, cell pellets were thawed, resuspended in buffer A (50 mM Na$_2$HPO$_4$/NaH$_2$PO$_4$ pH 7.5, 500 mM NaCl, 10% glycerol, 30 mM imidazole (3899.1, Roth), 0.2% Triton-X-100 (T8787, Sigma), 1 mM TCEP (HN95.3, Roth), cOmplete protease inhibitor (4693132001, Roche), 0.04 mg/mL Pefabloc SC (76307, Sigma), 1 mg/mL lysozyme (8259.3, Roth), 45 U/mL smDNAse (MPI for Biochemistry)) and incubated on a roller for 30 min. Cell lysate was sonicated for 15 min before removal of cell debris by centrifugation at 18,000 x *g* for 30 min. Clarified lysate was filtered using syringe filters (PVDF, 0.22 µm) and incubated with Ni-NTA agarose (30250, Qiagen) on a roller for 1 h at 4 °C. The beads were transferred to a gravity flow column and washed with 15 CV buffer B (50 mM Na$_2$HPO$_4$/NaH$_2$PO$_4$ pH 7.5, 500 mM NaCl, 10% glycerol, 30 mM imidazole (3899.1, Roth)) before elution in 2 CV buffer C (50 mM Na$_2$HPO$_4$/NaH$_2$PO$_4$ pH 7.5, 500 mM NaCl, 10% glycerol, 300 mM imidazole (3899.1, Roth)). The His-tag was cleaved by the addition of His-tagged TEV protease (1 mg/L culture) (ref. 37). The protein was dialyzed against buffer D (20 mM HEPES/KOH pH 7.5, 100 mM KCl) overnight. The next day, cleaved protein was passed through Ni-NTA agarose (30250, Qiagen) to remove His-tagged TEV protease, residual uncleaved protein and the His-Tag. Flow-through was concentrated to 1 mL and loaded on a Superdex 200 Increase 10/300 GL column (28990944, Cytiva) equilibrated in buffer E (20 mM HEPES/KOH pH 7.5, 100 mM KCl, 10% glycerol, 1 mM TCEP (HN95.3, Roth)). Eluted protein was concentrated with 3 kDa cut-off Amicon ultra centrifugal filters (UFC8003, Merck) before aliquoting, snap-freezing in liquid nitrogen and storage at −80 °C.

### In vitro HMCES-DPC generation

DPCs were generated between HMCES$^{SRAP}$, HMCES$^{SRAP}$-K48-Ub$^{[short]/[long]}$, HMCES$^{SRAP}$-K63-Ub$^{[short]/[long]}$ or HMCES$^{FL}$ and a 30nt Cy5-labeled forward oligonucleotide (5'-Cy5-CCCAAAAAAAAAAAdUAAAAAAAAAAAA CCC-3'), as previously described[37,38]. For HMCES$^{FL}$-DPCs final concentrations differed from published protocols: HMCES$^{FL}$ (13 µM), UDG (0.1 U/µL) (M0280L, NEB), DNA (1.25 µM). For all crosslinking reactions, incubation was shortened to 30 min at 37 °C. To form ssDNA-dsDNA junctions 1 µL complementary 15nt reverse oligonucleotide (5'-GGGTTTTTTTTTTTTT-3') (12 µM in nuclease-free H$_2$O) was annealed to all crosslinking reactions.

### HMCES$^{SRAP}$ Ubiquitylation using synthetic ubiquitin E3 ligases

A simplified Ubiquiton system[35], based on fusions of a complete ubiquitin instead of split-ubiquitin as a starting point, was used. In brief, HMCES$^{SRAP}$-Ub(G76V)−3C-FKBP-His6 was K48-poly-ubiquitylated in a reaction containing substrate (10 µM), ubiquitin (30 µM) (U6253, Merck), Ub$^1$-K48R (10 µM) (IMB gGmbH), His-Uba1 (50 nM) (refs. 51,52), Ubc7-His (E2) (4 µM) (refs. 51,52), His-FRB-E3[48] (10 µM) (IMB gGmbH), ATP (1 mM) (R0441, Thermo Scientific) and rapamycin (50 µM) (SEL-S1039, Biozol) in ubiquitylation buffer (40 mM HEPES/NaOH pH 7.4, 50 mM NaCl, 8 mM Mg(OAc)$_2$) for 6.5 h at 30 °C. K48-modified HMCES$^{SRAP}$ was separated from other reaction components by cleaving the dimerization tag using His-3C-protease (IMB gGmbH) at 4 °C overnight, reverse immobilized metal affinity chromatography (IMAC) and SEC (20 mM HEPES/KOH pH 7.8, 150 mM KCl, 2 mM MgCl$_2$, 1 mM TCEP (HN95.3, Roth), 10% glycerol).

HMCES$^{SRAP}$-Ub(G76V)−3C-FKBP-His6 was K63-poly-ubiquitylated in a reaction containing substrate (10 µM), ubiquitin (30 µM) (U6253, Merck), Ub$^1$-K63R (10 µM) (IMB gGmbH), His-Uba1 (50 nM) (refs. 51,52), His-Ubc13·Mms2 (E2) (2 µM) (refs. 51,52), His-FRB-L20-E3[63] (10 µM) (IMP gGmbH), ATP (1 mM) (R0441, Thermo Scientific) and rapamycin

(50 μM) (SEL-S1039, Biozol) in ubiquitylation buffer (40 mM HEPES/NaOH pH 7.4, 50 mM NaCl, 8 mM Mg(OAc)$_2$) for 2 h at 30 °C and purified as described above.

Ubiquitin mutants, His-Uba1, Ubc7-His, His-Ubc13 and Mms2 were purified as previously described[51,52]. His-FRB-E3[48] and His-FRB-L20-E3[63] were produced in *E. coli* and purified by IMAC followed by SEC (20 mM HEPES/NaOH pH 7.4, 150 mM NaCl, 10% glycerol, 1 mM DTT (D0632, Merck).

## In vitro SUMOylation and ubiquitylation of HMCES-DPCs

SUMOylation reactions were performed in 20 μL for 30 min at 37 °C, containing HMCES-DPC (125 nM), SUMO2 (1.250 μM), SAE1/UBA2 (100 nM) (NKM-ATGP3363, Hölzel), UBC9 (200 nM) and PIAS4 (125 nM). The reaction buffer comprised 20 mM HEPES/KOH pH 7.5, 110 mM KOAc, 5.32 mM MgCl$_2$, 2 mM Mg(OAc)$_2$, 0.05% TWEEN20 (P7949, Sigma), 0.2 mg/ml BSA (AM2616, Thermo Scientific), 1 mM TCEP (HN95.3, Roth), 2.5 mM ATP (R0441, Thermo Scientific). If no further reactions were carried out 5 μL reaction buffer were added and DPCs were either used in DPC cleavage assays or directly mixed with 4x LDS sample buffer (NP0007, Thermo Scientific) supplemented with 5% β-mercaptoethanol (β-ME) (M3148, Sigma), followed by boiling for 1 min at 95 °C prior to SDS-PAGE analysis. For subsequent ubiquitylation, 5 μL ubiquitin master mix were added, and reactions were incubated for 30 min at 37 °C. The ubiquitin master mix contained mono-ubiquitin (1 μM), UBE1 (100 nM) (182UB101, Lifesensors), RNF4 (200 nM) and UBE2D3 (200 nM). DPCs were either used in DPC cleavage assays or directly mixed with 4x LDS sample buffer (NP0007, Thermo Scientific) supplemented with 5% β-ME (M3148, Sigma), followed by boiling for 1 min at 95 °C prior to SDS-PAGE analysis. Samples were resolved on SDS-PAGE gels (12% Bis-Tris BOLT, Thermo Scientific) using MOPS buffer. Gels were scanned using a BioRad Chemidoc MP system with appropriate filter settings for Cy5 fluorescence. Gels were subsequently analyzed by immunoblotting using anti-K48-Ub (D9D5) (1:500) (8081S, Cell Signaling) and anti-K63-Ub (D7A11) (1:500) (5621S, Cell Signaling) antibodies.

For analysis of SUMOylated and ubiquitylated HMCES[FL]-DPC by mass spectrometry (MS), reactions were scaled up to 50 μL, ubiquitin concentration was increased (5 μM) and incubation time for ubiquitylation was extended (1 h at 37 °C). Reactions were stopped by addition of 4x LDS sample buffer (NP0007, Thermo Scientific) supplemented with 5% β-ME (M3148, Sigma). Samples were stored at −20 °C until MS analysis.

## DPC cleavage assay

DPC cleavage by SPRTN was assessed in 10 μL reactions at 30 °C for 1 h, containing SPRTN (WT or variants, as indicated – concentrations ranging from 0.1–100 nM), DPC or free DNA (10 nM) with or without FANCJ (100 nM) and with or without free K48-linked tetra-ubiquitin (Ub$^4$) (SI4804, Lifesensors) or K63-Ub$^4$ (SI6304, Lifesensors) (400 nM, referring to concentrations of individual ubiquitin moieties). The reaction buffer comprised 17.1 mM HEPES/KOH pH 7.5, 85.6 mM KCl, 3.1% glycerol, 5 mM TCEP (HN95.3, Roth), 2.1 mM MgCl$_2$, 0.12 mg/ml BSA (AM2616, Thermo Scientific) and 1 mM ATP (R0441, Thermo Scientific). Reactions were stopped with 4x LDS sample buffer (NP0007, Thermo Scientific) supplemented with 5% β-ME (M3148, Sigma) and boiling for 1 min at 95 °C. Samples were resolved on SDS-PAGE gels (12% Bis-Tris BOLT, Thermo Scientific) using MOPS buffer. Gels were scanned using a BioRad Chemidoc MP system with appropriate filter settings for Cy5 fluorescence. DPC cleavage was quantified using ImageJ (v1.54 f), by dividing the amount of cleaved DPCs by the total amount of DPC (cleaved + uncleaved). For Cleaved DPC*, the sum of cleavage fragment signals and the corresponding signal for free DNA was calculated minus free DNA signals inferred from control DPC reactions.

## Cellular SPRTN autocleavage assays

For cellular SPRTN autocleavage assays, $1 \times 10^6$ cells were seeded in 6-well plates. 24 h later 4 μL siRNA (20 μM) and 20 μL Lipofectamine RNAiMAX Transfection Reagent (13778075, Thermo Scientific) were each diluted in 200 μL Opti-MEM serum-free medium. Following a 5 min incubation, siRNA and Lipofectamine RNAiMAX Transfection Reagent dilutions were mixed. After an additional 15 min, the transfection mix was added to cells. 24 h after transfection, each well was split in 4 wells of a 12-well plate. The next morning, cells were treated with 5-azadC (10 μM) (A3656, Sigma) for 2, 4 or 8 h or left untreated for each transfected siRNA. At desired time points, cells were directly lysed in 1x LDS (NP0007, Thermo Scientific) and boiled for 20 min at 95 °C. Samples were resolved on SDS-PAGE gels (4-12% Bis-Tris NuPAGE, Thermo Scientific) using MOPS buffer and subsequently immunoblotted using anti-DNMT1 antibody (1:1000) (#5032, Cell Signaling), anti-SPRTN antibody (1:500) (6F2) (ref. 26), anti-RNF4 antibody (1:500) (AF7964, R&D systems) and anti-Vinculin antibody (1:1000) (sc-73614, Santa Cruz Biotechnology). The following siRNAs (Horizon Discovery) were used: siCTRL (Control pool, D-001810-10-20), siRNF4 (SMARTpool,L-006557-00-0005).

## Purification of x-linked proteins (PxP)

For PxP experiments, $7.5 \times 10^5$ cells were seeded in 6-cm dishes, and thymidine-containing media (2 mM) (T9250, Sigma) was added after 8 h. After approximately 16 h, thymidine media was removed, and cells were washed twice with PBS and released into normal media for 9 h, before thymidine media was re-added and expression of YFP-SPRTN-Strep-tag variants was induced with DOX (1 μg/mL) (D9891, Sigma). After another 16 h in thymidine media, cells were washed twice with PBS and released into normal media for 2 h before adding fresh media containing 5-azadC (10 μM) (A3656, Sigma). After a 30 min incubation, 5-azadC containing media was removed, cells were washed twice with PBS and recovery was allowed for 2 h. Cells were scraped on ice at indicated time points and cell pellets were stored at −80 °C. PxP to isolate DNMT1-DPCs was performed as previously described[9,42]. In brief, 10 μL of each cell suspension were directly lysed in 1x LDS sample buffer (NP0007, Thermo Scientific) to serve as input samples before plug casting. $1.5 \times 10^6$ cells were embedded into low-melt agarose (1613111, Bio-Rad) plugs, extracted by PxP[9,42] and prepared for analysis by SDS-PAGE at the end of the protocol. Samples were resolved on SDS-PAGE gels (4-12% Bis-Tris NuPAGE, Thermo Scientific) using MOPS buffer and subsequently immunoblotted using anti-DNMT1 antibody (1:1000) (#5032, Cell Signaling), anti-SPRTN antibody (1:500) (6F2) (ref. 26), and anti-β-actin antibody (1:1000) (Sc47778, Santa Cruz Biotechnology).

## Cell viability

For cell proliferation assays, 1,000 cells were seeded per well (n = 8) in a 96-well plate, and cell numbers were recorded every 8 h for 3 days using Cytation 5 (BioTek) equipped with a 4x objective and the Gen5 software (ver. 3.14).

## Flow cytometry

Cells were labeled with EdU (10 μM) for 45 min. EdU staining was performed with the Click-iT EdU Alexa Fluor 488 Flow Cytometry Assay Kit (C10425, Thermo Scientific) following the manufacturer's protocol. Cells were next stained with 4′,6-diamidino-2-phenylindole (DAPI) (4 μg/mL) (62248, Thermo Scientific) and analyzed using the BD LSRFortessa cell analyzer (BD Biosciences) with the FACSDiva™ software (ver. 6.2). Figures were generated using FlowJo (ver. 10.10).

## Microscopy

Cells grown on a cover glass were washed once with PBS, fixed with paraformaldehyde (4%) (P6148, Sigma) in PBS for 10 min, and permeabilized with 0.2% Triton X-100 (T8787, Sigma) in PBS for 10 min.

After washing with PBS, cells were stained with DAPI (1 μg/mL) (62248, Thermo Scientific) in PBS for 10 min, and the cover glass was mounted with ProLong Glass Antifade Mountant (P36980, Thermo Scientific). Images were captured using a Zeiss Axio Observer 7 equipped with Apotome 3 and the Axiocam 820 camera. At least 300 DAPI-stained nuclei were scored manually for the presence of micronuclei or chromatin bridges by an observer blinded to sample identities. Statistical analysis was performed by two-way ANOVA with Dunnett's multiple comparison test in GraphPad Prism (ver. 10.3.0).

### Identification of ubiquitin-linkages by quantitative mass spectrometry analysis

For SUMO-targeted ubiquitylated HMCES[FL]-DPCs (4 biological replicates per condition), reactions were terminated by boiling samples at 70 °C. Synthetically ubiquitylated HMCES[SRAP] (3 biological replicates per condition) was directly used for mass spectrometry measurements. For quantitative mass spectrometry analysis, samples were subsequently cleaned-up using the paramagnetic-based SP3 technology as described previously[53]. Briefly, 100 μg of freshly pre-equilibrated SP3 beads (45152105050250, GE Healthcare), were added to 20 μL of samples. Purification of total proteins from in vitro reactions was next completed through three-rounds of 80% (v/v) ethanol-solvation of the SP3-sample mixture at room temperature. The resulting purified proteins were then subjected to trypsin digestion (1 μg) in 50 mM ammonium bicarbonate pH 8.0 for 16 h at 37 °C. Digested peptides were acidified using trifluoroacetic acid and desalted on reverse-phase C18 StageTips for MS analysis. Eluted samples were analyzed on a quadrupole Orbitrap mass spectrometer Exploris 480 (Thermo Scientific) equipped with a UHPLC EASY-nLC 1200 system (Thermo Scientific). Samples were loaded onto a C18 reversed-phase column (55 cm length, 75 mm inner diameter, packed in-house with ReproSil-Pur 120 C18-AQ 1.9-μm beads) (r119.aq, Dr. Maisch GmbH) and eluted with a gradient from 2.4 to 32% Acetonitrile.

The mass spectrometer was operated in data-dependent mode, automatically switching between MS and MS2 acquisition. Survey full scan MS spectra (m/z 300–1,650; resolution: 60,000; target value: $3 \times 10^6$; maximum injection time: 28 ms) were acquired in the Orbitrap. The 15 most intense precursor ions were sequentially isolated, fragmented by higher energy C-trap dissociation (HCD) and scanned in the Orbitrap mass analyzer (normalized collision energy: 30%; resolution: 15,000; target value: $1 \times 10^5$; maximum injection time: 40 ms; isolation window: 1.4 m/z (LFQ run)). Precursor ions with unassigned charge states, as well as with charge states of +1 or higher than +6, were excluded from fragmentation. Precursor ions already selected for fragmentation were dynamically excluded for 25 s.

Peptide identification: Raw data files were analyzed using Max-Quant (development version 1.5.2.8). Parent ion and MS2 spectra were searched against a database containing 98,566 human protein sequences obtained from UniProtKB (April 2018 release) using the Andromeda search engine. Spectra were searched with a mass tolerance of 6 ppm in MS mode, 20 ppm in HCD MS2 mode and strict trypsin specificity, allowing up to three miscleavages. Protein N-terminal acetylation and methionine oxidation were searched as variable modifications. The dataset was filtered based on posterior error probability (PEP) to arrive at a false discovery rate (FDR) of less than 1% estimated using a target-decoy approach.

### ICP-OES measurements

SPRTN samples in storage buffer and expression media (TB) as control were digested using an *Anton Paar* Multiwave 5000 microwave. For this, 160 μL of each sample (corresponding to 0.95 mg protein) were placed into PTFE digestion vessels. To this, 1 mL HNO₃ (69%) (450041 M, VWR) was added. The used digestion program was: 5 min ramp up to 180 °C, then 10 min at 200 °C and 15 min at 220 °C. After digestion, samples were allowed to cool to room temperature before

being diluted to final volumes (10 mL) with ultrapure water (type 1, 18.2 MΩ·cm at 25 °C) for Inductively Coupled Plasma Optical Emission Spectrometry (ICP-OES) analysis. Blank samples were treated analogously.

ICP-OES was performed on a Varian Vista RL instrument operating in radial mode to determine the concentrations of Co, Fe, Mn and Zn. Calibration standards were prepared in HNO₃ (2%) by diluting a certified multi-element ICP standard (1.09492, Merck) containing the elements of interest to obtain a 4-point linear calibration curve ranging from 0 μg/mL to 4 μg/mL. ICP-OES operating conditions were set as follows: plasma power at 1.25 kW, plasma gas at 13.5 L/min, nebulizer pressure at 170 kPa, auxiliary gas flow rate at 1.5 L/min with three replicates per measurement cycle, which were automatically averaged. The following emission lines were selected for Co at 230.786, 231.160, 237.863 and 258,033 nm, Fe at 234.350, 238.204, 258.588 and 259.940 nm, Mn at 257.610, 259.372 and 293.931 nm and Zn at 202.548, 206.200 and 213.857 nm. Quality control was ensured by analyzing blanks within the sequence and a certified reference material alongside the samples, with recoveries within acceptable limits. See Supplementary Table 1 for results.

### Protein structure predictions

To prepare the crystal structure (PDB: 6mdx) for MD simulations, Swiss Model[54] was employed to model the two missing residues, to adjust the modified amino acids to their natural counterpart and to remove the ligands from crystallization. From the structures generated with Swiss Model, we took the one that was the closest to the crystal structure with an RMSD of 0.082 Å. Structures for SprT-BR (SPRTN[aa28-245]) and SprT-BR-ubiquitin were predicted using ColabFold[39,55,56] and AlphaFold2[57] using alphafold2_ptm. Figures were generated using PyMOL (ver. 3.0.3).

### Molecular dynamics simulations

Starting structures for SprT and SprT-Ubiquitin were generated as described above, in both cases the top-ranked model (Rank_1) was selected for further analysis. Predictions for SPRTN variants (L38S, L99S) were generated analogously. In case of SprT-ubiquitin complexes, the interface predicted for the WT enzyme was also used for SPRTN variants. Starting with these predicted structures, two Zn²⁺ ions were added based on their binding sites in the crystal structure. Subsequently, hydrogen atoms were added to these structures as well as to the model of the crystal structure employing the H++ server, which determines protonation states based on continuum electrostatics[58]. Specifically, H++ employs the Poisson–Boltzmann equation to estimate the pKa of the ionizable residues in a macromolecule and assigns the protonation states accordingly[59]. This way, the model for SprT and for the crystal structure consisted of the same atoms and of amino acids in the same protonation states. Pdb-files of the SprT domain (WT and variants L38S, L99S) are provided as supplementary data (Supplementary Data 1-3). Figures and movies were generated using PyMOL (ver. 3.0.3).

For setting-up the system for the simulations, we employed AmberTools20[60]. The structures were placed in rectangular simulation boxes with a minimum distance of 15 Å between the solute and the boundaries of the box. The boxes were filled with water and NaCl was added to neutralize the system and to achieve a physiological concentration of around 150 mM NaCl leading to system sizes of about 72.324 to 95.881 atoms for SprT and the SprT-Ub¹ complex, respectively.

The force field parameters for the proteins were taken from the Amber ff19SB force field[61] and the OPC water model was used[62]. After conversion of the topology and coordinate files to gromacs with parmed from AmberTools20, the parameters for NaCl were replaced by the ones from ref. [63] and for Zn²⁺ by the parameters from ref. [64].

MD simulations were performed using the Gromacs simulation package[65], version 2024. Initially, the energy of the systems was

minimized using the steepest descent algorithm. Subsequently, the systems were equilibrated for 1 ns, first in the NVT and then in the NPT ensemble. For the production run, we performed 400 ns long simulations employing the velocity-rescaling thermostat with a stochastic term and a time constant of 0.1 ps and isotropic Parrinello-Rahman pressure coupling with a time constant of 5.0 ps. Each production was repeated three times with random velocities. We used clustering analysis with Gromacs for the production runs based on RMSD to group similar conformations, allowing us to identify the dominant structural states and to calculate the radius of gyration (Rg) of the clusters. In this analysis, 100 ns were discarded for equilibration. For the clustering, we employed the Gromacs bulit-in tool gmx cluster using the Daura clustering algorithm with an RMSD cut-off of 0.5 nm for all atoms of the protein. The algorithm identifies neighbors for each structure within the specified cut-off, selects the structure with the most neighbors as the first cluster center, and groups it with its neighbors. These are then removed from the pool, and the process repeats until all structures are clustered[66]. The numbers of clusters for the simulated systems and the fraction of structures in the three most populated clusters are summarized in Supplementary Table 2.

To calculate the binding affinity, additional simulations were performed. First 1 ns of NVT and NPT simulations with stronger restraints of 2.1 kcal/mol Å$^2$. Then production runs with weak restraints of 0.1 kcal/mol. We used representative structures from the largest cluster of the unrestrained simulations as starting structures. The productions were performed for 4 ns neglecting the first ns for equilibration and structures were sampled every 10 ps. Each simulation was repeated three times. The end-point free energy calculations were performed using the MMPBSA program from the Amber package[67] using the gmx_MMPBSA tool[68]. Water molecules and ions were removed and the trajectories were re-evaluated using the mbondi3 radii and parameters from ref. 69 denoted as igb=8 in Amber. To account for hydrophobic solvation, we used a surface area-dependent tension model with surface tension coefficient $\gamma = 0.005$ kcal/mol Å$^2$. No conformational changes were considered. From the alanine scanning procedure, we obtained the contribution of the two mutations to the binding free energy. The energy contribution of the mutation was calculated by cutting all atoms after the Cβ-atom of this residue. This procedure was performed for the simulations of the WT and the simulations of the mutated complexes. The difference yielded the binding energy contributions of the L38S and the L99S mutations.

## NMR spectroscopy
All NMR samples (uniformly labeled $^{13}$C,$^{15}$N-ZBD-BR, $^{15}$N-SprT-BR, $^{15}$N-SprT-BR-L99S, $^{15}$N-SprT-BR-W36G, $^{15}$N-SprT-BR-W58G, $^{15}$N-ZBD) were prepared at concentrations of 100 μM and 250 μM in 20 mM HEPES/KOH pH 7.2, 150 mM KCl with 10% D$_2$O, as lock signal. All NMR experiments were recorded at 308 K on a 600 MHz Bruker Avance NMR spectrometer, equipped with a cryogenic triple-resonance gradient probe. NMR spectra were processed using NMRPIPE[70] or TOPSPIN3.7 (Bruker) and analyzed using NMRFAM-SPARKY[71]. Using the previous backbone resonance assignments for ZBD-BR from ref. 21, aromatic resonances were further assigned using 2D CBHD, CBHE, aromatic $^1$H,$^{13}$C-HSQC and 3D $^{15}$N/$^{13}$C-edited NOESY experiments. Trp ε1 resonances for the protease domain were assigned by mutation (W36G or W58G), while W68 resonance was assigned by exclusion. For 2D $^1$H,$^{15}$N-HSQC comparisons, 100 μM SPRTN was mixed with 500 μM Ub$^1$ or Ub$^1$-I44A (5x molar excess) and / or 200 μM dsDNA (2x molar excess) (fwd: 5′-CCTTGCTAGGACATC-3′ + rev: 5′-GATGTCCTAGCAAGG-3′, annealed to dsDNA) accordingly. Chemical shift perturbations (CSP) values were calculated based as $\Delta\delta_{HN,N} = \sqrt{\Delta\delta_{HN}{}^2 + \left(\frac{\Delta\delta_N}{R_{scale}}\right)^2}$, where $R_{scale} = 6.5$ was applied as suggested previously[72].

## Reporting summary
Further information on research design is available in the Nature Portfolio Reporting Summary linked to this article.

## Data availability
Mass spectrometry data reported in this manuscript have been deposited to the ProteomeXchange Consortium (www.proteomexchange.org) via the Proteomics Identification Database (PRIDE) partner repository with the dataset identifier PXD063921. All remaining data supporting the findings of this study are available from the corresponding author upon request. Source data are provided with this paper.

## Code availability
All conformers from the MD simulation trajectories are available from the corresponding author upon request.

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

## Acknowledgements

We thank B. Schulman for providing a plasmid for ubiquitin. We gratefully acknowledge Jiaxuan Chen from the IMB Proteomics Core Facility in Mainz for help with mass spectrometry experiments. S.D. and P.W. are supported by the International Max-Planck Research School for Molecules of Life. We are grateful to Sam Asami and Gerd Gemmecker for help with NMR measurements at the Bavarian NMR center. We thank Dr. Shar-yin N. Huang at the National Cancer Institute for her technical support. Research in the lab of J.S. is funded by European Research Council (ERC StG 801750 DNAProteinCrosslinks, ERC CoG 101124695 DECONSTRUCT), the Alfried-Krupp von Bohlen und Halbach-Stiftung, European Molecular Biology Organization (YIP4644), a Vallee Foundation Scholarship, and Deutsche Forschungsgemeinschaft (Project ID 213249687 - SFB 1064). H.D.U. acknowledges funding by the European Research Council (ERC AdG 101140447). We acknowledge funding by the Deutsche Forschungsgemeinschaft (J.S., H.D.U., and P.B.: Project-ID 393547839 – SFB 1361; M.S. and L.J.D: Project-ID 325871075 – SFB1309). The authors acknowledge the scientific support and HPC resources provided by the Erlangen National High Performance Computing Center (NHR@FAU) of the Friedrich-Alexander-Universität Erlangen-Nürnberg (FAU) under the NHR project b119ee and the resources on the LiCCA HPC cluster of the University of Augsburg, co-funded by the Deutsche Forschungsgemeinschaft under Project-ID 499211671).

## Author contributions

Conceptualization: S.D. and J.S. Investigation: S.D., D.S.S., D.Y., P.W., M.J.G., Y.M., Y.J.M., and C.R. NMR: H.S.K. MD-simulations: C.W., A.C.M., and N.S. Mass spectrometry: A.S.R. ICO-OES: S.M.G.T. Writing – Original draft: S.D. and J.S. Writing – Review & Editing: S.D. and J.S. with input from all authors. Funding Acquisition and Supervision: J.S., L.J.D., P.B., H.D.U., M.S., Y.J.M., and N.S.

## Funding

## Competing interests

The authors declare no competing interests.
