## [Transparent Peer Review file · Nature Communications]

Allosteric activation of the SPRTN protease by ubiquitin maintains genome stability

Corresponding Author: Professor Julian Stinglele

Version 0:

Reviewer comments:

Reviewer #1

(Remarks to the Author)

SPRTN is a specialized protease that plays a central role in the repair of DNA protein crosslinks (DPCs), particularly during DNA replication. In the absence of repair, DPCs obstruct DNA replication and transcription, leading to cellular lethality, or genome instability when the DPC removal functions less efficiently. SPRTN has a ubiquitin (Ub) binding (UBZ) domain, and ubiquitination of protein substrates is thought to be important for SPRTN activity. Unexpectedly, it was shown previously that the UBZ domain is not essential, suggesting that there might be additional ways how ubiquitin regulates SPRTN.

In the current study, the authors provide insights into the activation of the SPRTN protease activity by ubiquitin. The authors use purified proteins and synthetic DPC substrates in very impressive reconstitution biochemistry experiments. It is shown that SPRTN is activated by the ubiquitination of DPCs, independent of the UBZ domain. Using modeling, the authors identified an interface on the SPRTN protease domain (termed USD interface), which mediates interaction with Ub, and the interaction is proposed to facilitate open/active conformation of the SPRTN protease domain, explaining its activation. Some residues (e.g., L99) in the USD interface are identified, and mutations (e.g. L99S) are tested in vitro as well as in cells in complementation assays, where defects in the ubiquitin-binding interface of SPRTN are shown to lead to genomic instability and cell cycle defects. Finally, it is shown that the newly identified interface functions alongside the canonical UBZ domain, and both are necessary for full activity. In my opinion, this is a well done study that will be suitable for publication pending some additional control experiments and clarifications.

Specific comments:

- Figure 1: It is important where the Ub moiety is on the DPC? Fig. 1d: does Ub in solution boost SPRTN activity (in trans)? I did not see this control. A curiosity (not necessary experiment): have the authors tried whether N- or C-terminal fusions of Ub with the DPC work the same as mono-Ub added by the Ub machinery?

- Fig. 2: Are the interaction patches conserved? Please show a multiple sequence alignment.

- Figure 3 (and Fig. 4f): The effects of the mutations in the SPRTN interface are rather small. To better functionally validate complex formation based on the model, the authors could try combining several mutations in SPRTN and/or mutate also reciprocally Ub. The concern is that the key interface is somewhere else. Can the effect of the mutations be observed in pulldown experiments with full-length proteins in vitro or in cells?

- Fig. 4: Please quantitate what fraction of SPRTN is modified by Ubiquitin. Does it explain the decreased efficacy of DPC cleavage in the fully reconstituted system? Regarding "absent (L99S)" in the text referring to panel 4f, the products are still visible, as also apparent in the quantitation. Please rephrase.

- A discussion point: The authors introduce that it is not clear how SPRTN "knows" which substrates to cleave, and why it does not cleave non-DPC DNA-associated chromatin proteins. It is shown here that SPRTN activity depends on Ubiquitination. How do the Sumo/Ub machinery proteins "know" what to modify?

- In a previous study (PMID: 35469923), it was shown that Ub stimulates the p97 enzyme that is thought to unfold proteins before SPRTN proteolytic action. While the current manuscript is not in conflict with the published work, it should be discussed.

Reviewer #2

(Remarks to the Author)

In this study, the authors have conducted a comprehensive biochemical analyses addressing the role of ubiquitin in SPRTN activation and DPC proteolysis. Using in vitro reconstituted DPCs systems, molecular dynamics simulations, NMR, and cellular assays, the authors show that ubiquitin activates SPRTN protease activity and DPC cleavage. This activation does not require SPRTN UBZ domain, but is dependent on USD, an ubiquitin binding interface in the SprT domain. This finding is significant as it is not known how DPC ubiquitylation regulates SPRTN-mediated DPC proteolysis in the field. Overall, well defined comprehensive analysis. Identification of USD in SPRTN, and the role of ubiquitin in SPRTN-mediated DPC proteolysis is significant and well written manuscript with detailed methodology.

Queries:

1. In fig 1d and 1e, using a in-vitro reconstituted ubiquitylated DPC system the authors show that SPRTN cleaves ubiquitylated DPCs more efficiently compared to the unmodified DPCs. FANCI is used in this assay to unfold the protein adduct making the substrate available for SPRTN protease. Will FANCI activity on ubiquitylated DPCs influence or effect DPC cleavage by SPRTN? Does FANCI unfold unmodified and modified (short vs long chained ub-DPCs) DPCs to similar extent allowing for DPC cleavage by SPRTN?

2. In Fig 4g, it is stated that DNMT1-DPC cleavage is reduced in cells expressing SPRTN L38S. However, the effect of SPRTN E112Q, L38S, and L99S on DPC cleavage (red dots indicated in the figure) appears to be the same in the 2h chase samples. Further, DNMT1-DPCs in 0h chase samples (lane 11) are dramatically reduced (DNMT1 band indicated in the top blot) in cells expressing L99S compared to WT (lane 2), E112Q (lane 5), and L38S (Lane 8) samples. Are DNMT1-DPCs processed by alternate pathways in cells expressing SPRTN L99S compared to E112Q or L38S? These effects need to be clarified and explained better.

Reviewer #3

(Remarks to the Author)

This manuscript by Durauer and coauthors reported an interesting finding that ubiquitin can bind to the SPRTN catalytic domain and allosterically activate its protease activity in cleaving DPCs. They identified and confirmed the site of Ub binding in the SPRTN's catalytic domain, i.e. the USD interface, using NMR. In an in vitro system, the stimulation of SPRTN's activity in cleaving DPC was demonstrated using artificially and enzymatically installed polyubiquitin chains. The residues at SPRTN's USP interface were interrogated using site-directed mutagenesis. Finally, the effect of this interaction was studied in cellular systems and the results in general agree with a role of ubiquitin in stimulating SPRTN catalytic activity through interaction with its USD interface. This is a thorough study that unveiled an important regulatory mechanism of SPRTN's activity and DPC repair. It is of high interest to researchers in both the DNA repair and ubiquitin fields. Despite the strength of the study, there are several important questions that need to be addressed (listed below).

Do the authors know that the polyubiquitin chains formed using the modified Ubiquitin strategy are only conjugated to the fused Ub moiety, but not the HMCESSRAP domain? This was assumed, but need to be experimentally confirmed.

In the in vitro cleavage of DPC experiments, two major cleavage products with well-defined bands were detected. What is the molecular nature of these cleavage products. Knowing this is important considering that different mutants of SPRTN showed distinct effects on the formation of these two products.

Have the authors added free Ub to the assays described in Figure 1 d and e, particularly when unmodified HMCESSRAP was used, to determine the effect on DPC cleavage? If the concentration of free Ub is high enough, a stimulation of SPRTN's protease activity is expected.

As seen in Figure 2m, the effect of Ub is rather small on SprT-BR with minimal chemical shift changes. However, DNA binding greatly enhanced the effect of Ub on SprT conformation. A model should be proposed to explain this interesting observation.

As shown in Figure 3, the effect of L99S on DPC cleavage is less pronounced when the total DPC cleavage products are compared, while the amount of the smaller cleavage product DPC* was significantly reduced by the L99S mutation. This needs to be explained by knowing the molecular nature of the two DPC cleavage products.

The DPC cleavage results obtained using DPC with either K48 or K63-linked polyUb chains showed little differences. This

is a somewhat unexpected observation considering that the K48 chain is more compact, which may obstruct the binding of the Ile44 patch in the polyUb chain to the USD in the SPRTN's catalytic core domain.

In Figure 4h, the effect of L99S mutation on the cleavage of DPC was minimal in the presence of enzymatically installed polyubiquitin chains. This is different from the artificially installed polyUb chains. While it is known that the polyubiquitin chains are installed at different sites on HMCES, it is still difficult to rationalize this observation.

Reviewer #4

(Remarks to the Author)

This paper explores the relationship between DNA-protein crosslinks (DPC) ubiquitylation and SPRTN protease activation associated with their degradation by implementing a combination of experimental and computational techniques to understand the regulatory role played by ubiquitylation in SPRTN mediated DNA repair mechanisms. Results from in-vitro assays of ubiquitylated DPC's treated with a truncated SPRTN (SprT-BR), combined with structure prediction methods reveal a novel open conformation of SPRTN where a hydrophobic region (L38, L99) in SprT-BR interacts with ubiquitin. MD simulations and NMR experiments of SprT-BR mutants further confirm the role of this interactions, leading authors to propose a model in which the open conformation of SPRTN is stabilized by simultaneous binding of DNA and ubiquitin binding. Authors finally extend their results to investigate the role of ubiquitin-dependent activation of truncated SPRTN variants as seen in Ruijs-Aalfs (RJALFS) syndrome patients.

The authors novelty in this paper lies in the identification of a ubiquitin-binding interface at the SprT domain (USD). However, at the introduction they also mention a similar motif interacting with ubiquitin-binding region (MIU) which is not clearly explained in the paper or in the corresponding citation that mentions MIU (27). Clear distinction between MIU and USD would further strengthen the papers findings.

For the computational methodology, authors should mention how h++ provides an estimation of the protonation states of titratable residues. Authors should also specify the number of clusters and exact criteria for such for computational reproducibility, such as the RMSD of all atoms or backbone CA atoms. MD simulations of SPRT mutants can highlight experimental highlights further and provide molecular insights into the disruptive effects of L38S.

Version 1:

Reviewer comments:

Reviewer #1

(Remarks to the Author)

The authors have clarified all the points I raised during the review by either adding additional data or by discussion. The revised manuscript is of a high quality and I recommend acceptance.

Reviewer #2

(Remarks to the Author)

Manuscript by Durauer et al. is a comprehensive biochemical study delineating the role of ubiquitin in SPRTN activation and DPC proteolysis. The authors have successfully addressed the reviewers queries. The Identification of a ubiquitin-binding interface at the SprT domain (USD) in SPRTN, and the role of ubiquitin in SPRTN-mediated DPC proteolysis is significant. The revised manuscript is suitable for publication.

Reviewer #3

(Remarks to the Author)

The authors have addressed the reviewer's questions satisfactorily.

Reviewer #4

(Remarks to the Author)

Thank you for taking into account the previous suggestions.

Reviewer Comments (reproduced in their entirety, our responses in black, our numbering of comments added for reviewer #1, #3 and #4)

Reviewer #1 (Remarks to the Author)

SPRTN is a specialized protease that plays a central role in the repair of DNA protein crosslinks (DPCs), particularly during DNA replication. In the absence of repair, DPCs obstruct DNA replication and transcription, leading to cellular lethality, or genome instability when the DPC removal functions less efficiently. SPRTN has a ubiquitin (Ub) binding (UBZ) domain, and ubiquitination of protein substrates is thought to be important for SPRTN activity. Unexpectedly, it was shown previously that the UBZ domain is not essential, suggesting that there might be additional ways how ubiquitin regulates SPRTN.

In the current study, the authors provide insights into the activation of the SPRTN protease activity by ubiquitin. The authors use purified proteins and synthetic DPC substrates in very impressive reconstitution biochemistry experiments. It is shown that SPRTN is activated by the ubiquitination of DPCs, independent of the UBZ domain. Using modeling, the authors identified an interface on the SPRTN protease domain (termed USD interface), which mediates interaction with Ub, and the interaction is proposed to facilitate open/active conformation of the SPRTN protease domain, explaining its activation. Some residues (e.g., L99) in the USD interface are identified, and mutations (e.g. L99S) are tested in vitro as well as in cells in complementation assays, where defects in the ubiquitin-binding interface of SPRTN are shown to lead to genomic instability and cell cycle defects. Finally, it is shown that the newly identified interface functions alongside the canonical UBZ domain, and both are necessary for full activity. In my opinion, this is a well done study that will be suitable for publication pending some additional control experiments and clarifications.

We thank the reviewer for their supportive comments.

Specific comments:

- Figure 1:

1. It is important where the Ub moiety is on the DPC?

While DPC ubiquitylation stimulates SPRTN activity in all our experimental setups, we observed differences in SPRTN activity between the synthetic ubiquitylation system (Ub is only attached to the C-terminal ubiquitin-tag of the DPC) and the enzymatic SUMO-targeted ubiquitylation system (Ub is attached to several lysine residues on SUMO and the DPC). This indicates that the position of the ubiquitylation can influence the degree of activation (see Discussion, lines 350-358).

2. Fig. 1d: does Ub in solution boost SPRTN activity (in trans)? I did not see this control.

Following the reviewer's suggestion, we have tested whether the addition of free K48- and K63-linked tetra-ubiquitin chains (Ub⁴) stimulates the cleavage of HMCES^{SRAP}-DPCs. Both types of chains increased DPC cleavage and resulted in the formation of smaller cleavage products (Cleaved DPC*) (New Extended Data Fig. 1b; compare lanes 5 and 9 (K48), and lanes 15 and 19 (K63)). These effects were reduced in USD mutant variants (Fig. R1). However, the stimulating effect of free ubiquitin chains was significantly lower than the one of direct DPC ubiquitylation.

New Extended Data Fig. 1b. Free ubiquitin chains boost DPC cleavage by SPRTN. (b) HMCES^{SRAP}-DPCs (10 nM) were incubated alone or in the presence of FANCJ (100 nM), K48-tetra-ubiquitin (Ub⁴) or K63-Ub⁴ (400 nM, referring to the concentration of individual ubiquitin moieties) and indicated concentrations of SPRTN (1-100 nM) for 1 h at 30°C. Quantification: bar graphs represent the mean ± SD of three independent experiments. Values for cleavage of unmodified HMCES^{SRAP}-DPC are the same as in Fig. 1e.

Fig. R1: HMCES^{SRAP}-DPCs (10 nM) were incubated alone or in the presence of FANCJ (100 nM) and indicated variants of SPRTN (100 nM), with or without indicated concentrations of K63-tetra-ubiquitin (Ub⁴) (0.4 or 1.6 μM, referring to the concentration of individual ubiquitin moieties) for 1 h at 30°C. Quantification: bar graphs represent the mean ± SD of three independent experiments.

3. A curiosity (not necessary experiment): have the authors tried whether N- or C-terminal fusions of Ub with the DPC work the same as mono-Ub added by the Ub machinery?

We have performed the suggested experiment from Fig. 1e using a HMCES^{SRAP} with a C-terminal linear tetra-ubiquitin fusion and compared cleavage by SPRTN to cleavage of an unmodified HMCES^{SRAP}-DPC (Fig. R2). Overall cleavage was increased and additional smaller cleavage products (Cleaved DPC*) appeared.

Fig. R2. HMCEs^{SRAP}-DPCs (10 nM) were incubated alone or in the presence of FANCJ (100 nM) and indicated concentrations of SPRTN (1-100 nM) for 1 h at 30°C.

- Fig. 2:

4. Are the interaction patches conserved? Please show a multiple sequence alignment.

We have included the requested multiple sequence alignment in the revised manuscript, showing conservation of the key USD residues Leu38 and Leu99 (New Extended Data Fig. 2f).

f

New Extended Data Fig. 2f. A novel ubiquitin binding interface at the SprT domain. (f) Schematic of SPRTN's domain structure with multiple sequence alignment highlighting key residues in the SprT domain in *H. sapiens*, *M. musculus*, *X. laevis*, *D. melanogaster* and *C. elegans* SPRTN homologues (L38 = light green, L99 = dark green).

- Figure 3 (and Fig. 4f):

5. The effects of the mutations in the SPRTN interface are rather small. To better functionally validate complex formation based on the model, the authors could try combining several mutations in SPRTN and/or mutate also reciprocally Ub. The concern is that the key interface is somewhere else.

We have conducted the suggested experiment. We cloned and purified a USD double mutant (L38S+L99S). The double mutant behaved indistinguishably from the L99S single mutant (New Extended Data Fig. 5a-b, compare lanes 6-8 and lanes 9-11). These data suggest that the L99S substitution abrogates Ub binding at the USD interface, which is consistent with our NMR experiments.

New Extended Data Fig. 5. The USD promotes cleavage of ubiquitylated DPC by SPRTN. (a-b) Indicated HMCES^{SRAP}-DPCs (10 nM) were incubated alone or in the presence of FANCJ (100 nM) and indicated concentrations (0.1-100 nM) and variants of SPRTN (WT, L99S, L38S+L99S) for 1 h at 30°C. Quantification: bar graphs represent the mean ± SD of three independent experiments.

Redacted

6. Can the effect of the mutations be observed in pulldown experiments with full-length proteins in vitro or in cells?

As suggested by the reviewer, we performed pull-down experiments using recombinant Flag-tagged SPRTN variants. SPRTN variants were incubated with recombinant K63-linked tetra-ubiquitin, followed by Flag-based pull-down to assess ubiquitin binding. To test the USD's role, we used SPRTN E112Q (a catalytically inactive variant, to prevent autocleavage) and SPRTN E112Q combined with the L99S substitution. We also analyzed a construct lacking the UBZ domain (SPRTN^{ΔUBZ}) and the SprT-BR construct. Ubiquitin binding was observed for the full-length protein and was reduced in constructs lacking the UBZ domain (Figure R3a-b). We observed no effect on ubiquitin binding in these assays upon substituting Leu99 in the USD interface, likely due to the low affinity nature of the USD-Ub interaction in the absence of DNA, as indicated by our NMR experiments.

Fig. R1. (a-b) Indicated recombinant Flag-SPRNT variants (1 μ M) were incubated with K63-tetra-ubiquitin (Ub⁴) (0.5 μ M, referring to the concentration of individual ubiquitin moieties) for 15 min on ice. Flag-SPRNT variants were pulled down (Flag-PD) together with bound ubiquitin using Anti-Flag beads. Reactions were analyzed by immunoblotting.

- Fig. 4:

7. Please quantitate what fraction of SPRNT is modified by Ubiquitin. Does it explain the decreased efficacy of DPC cleavage in the fully reconstituted system?

We assume the reviewer is referring to the fraction of “HMCES” modified by ubiquitin, not SPRNT. We attempted to quantify the fraction of ubiquitylated HMCES^{FL}-DPC. However, distinguishing between SUMOylated and ubiquitylated DPCs proved challenging. First, because not all SUMOylated DPCs are subsequently ubiquitylated, second because different species migrate at similar positions in the gel, and third, because highly modified SUMOylated and ubiquitylated DPCs are not resolved well by the SDS-PAGE gel. However, based on a qualitative analysis (Fig. R4a-b), we estimate that less than 10% of HMCES^{FL}-DPCs get ubiquitylated, thus explaining the smaller effect on overall cleavage in the fully reconstituted system (compared to the synthetic system, where 100% of DPCs are ubiquitylated).

Fig. R2. (a) SUMO-targeted ubiquitylated HMCES^{FL}-DPCs generated as described in Fig. 4a, separated by denaturing SDS-PAGE and immunoblotting. Panel shown in the manuscript in Fig. 4b. (b) Plots for signal intensities from the respective lanes (Lane #2, #4 and #7) of the SDS-PAGE Cy5 scan shown in (a). Peaks for free DNA, unmodified HMCES^{FL}-DPC, SUMO-modified HMCES^{FL}-DPC (red arrows) and SUMO/Ub-modified HMCES^{FL}-DPC (green arrows) are labeled.

8. Regarding "absent (L99S)" in the text referring to panel 4f, the products are still visible, as also apparent in the quantitation. Please rephrase.

We agree and rephrased this sentence.

9. - A discussion point: The authors introduce that it is not clear how SPRTN "knows" which substrates to cleave, and why it does not cleave non-DPC DNA-associated chromatin proteins. It is shown here that SPRTN activity depends on Ubiquitination. How do the Sumo/Ub machinery proteins "know" what to modify?

During transcription- and replication-coupled DPC repair, it is the collision between the crosslinked protein and the RNA polymerase and the replication fork, respectively, that triggers ubiquitylation of the protein adduct. However, how DPCs are sensed outside of replication and transcription by the SUMO system is not fully understood. The current view in the field is that DNA-bound SUMO-E3 ligases of the PIAS family SUMOylate proteins with long residence time on DNA, thereby acting as a 'molecular stopwatch' (e.g., Borgermann et al, EMBO J, 2020). The precise details of this process remain however elusive.

10. In a previous study (PMID: 35469923), it was shown that Ub stimulates the p97 enzyme that is thought to unfold proteins before SPRTN proteolytic action. While the current manuscript is not in conflict with the published work, it should be discussed.

We thank the reviewer for pointing this out. We now cite this study in the revised manuscript.

Reviewer #2 (Remarks to the Author)

In this study, the authors have conducted a comprehensive biochemical analyses addressing the role of ubiquitin in SPRTN activation and DPC proteolysis. Using in vitro reconstituted DPCs systems, molecular dynamics simulations, NMR, and cellular assays, the authors show that ubiquitin activates SPRTN protease activity and DPC cleavage. This activation does not require SPRTN UBZ domain, but is dependent on USD, an ubiquitin binding interface in the SprT domain. This finding is significant as it is not known how DPC ubiquitylation regulates SPRTN-mediated DPC proteolysis in the field. Overall, well defined comprehensive analysis. Identification of USD in SPRTN, and the role of ubiquitin in SPRTN-mediated DPC proteolysis is significant and well written manuscript with detailed methodology.

We thank the reviewer for their encouraging feedback.

Queries:

1. In fig 1d and 1e, using a in-vitro reconstituted ubiquitylated DPC system the authors show that SPRTN cleaves ubiquitylated DPCs more efficiently compared to the unmodified DPCs. FANCI is used in this assay to unfold the protein adduct making the substrate available for SPRTN protease. Will FANCI activity on ubiquitylated DPCs influence or effect DPC cleavage by SPRTN? Does FANCI unfold unmodified and modified (short vs long chained ub-DPCS) DPCs to similar extent allowing for DPC cleavage by SPRTN?

Redacted

Redacted

2. In Fig 4g, it is stated that DNMT1-DPC cleavage is reduced in cells expressing SPRTN L38S. However, the effect of SPRTN E112Q, L38S, and L99S on DPC cleavage (red dots indicated in the figure) appears to be the same in the 2h chase samples. Further, DNMT1-DPCs in 0h chase samples (lane 11) are dramatically reduced (DNMT1 band indicated in the top blot) in cells expressing L99S

compared to WT (lane 2), E112Q (lane 5), and L38S (Lane 8) samples. Are DNMT1-DPCs processed by alternate pathways in cells expressing SPRTN L99S compared to E112Q or L38S? These effects need to be clarified and explained better.

The effect mentioned by the reviewer was related to unequal loading (as evident by the β -Actin loading control in the original experiment). We have therefore repeated the experiment, and the revised manuscript now includes a new version of the experiment with equal loading, showing comparable levels of DNMT1-DPC formation in cells expressing SPRTN L99S (New Fig. 5g). We also rephrased the text to state that DPC cleavage was comparably reduced in cells expressing L38S and L99S SPRTN mutant variants.

New Fig. 5g. SUMO-targeted DPC ubiquitylation activates SPRTN. HeLa-TREx *SPRTN^{ΔC}* Flp-In cells complemented with indicated YFP-SPRTN^{FL}-Strep-tag variants were treated as depicted (top) with 5-azadC (10 μ M) and harvested at indicated time points. DNMT1-DPCs were isolated using PxP (middle, see Methods) and analyzed by immunoblotting (bottom).

Reviewer #3 (Remarks to the Author)

This manuscript by Durauer and coauthors reported an interesting finding that ubiquitin can bind to the SPRTN catalytic domain and allosterically activate its protease activity in cleaving DPCs. They identified and confirmed the site of Ub binding in the SPRTN's catalytic domain, i.e. the USD interface, using NMR. In an in vitro system, the stimulation of SPRTN's activity in cleaving DPC was demonstrated using artificially and enzymatically installed polyubiquitin chains. The residues at SPRTN's USP interface were interrogated using site-directed mutagenesis. Finally, the effect of this interaction was studied in cellular systems and the results in general agree with a role of ubiquitin in stimulating SPRTN catalytic activity through interaction with its USD interface. This is a thorough study that unveiled an important regulatory mechanism of SPRTN's activity and DPC repair. It is of high interest to researchers in both the DNA repair and ubiquitin fields. Despite the strength of the study, there are several important questions that need to be addressed (listed below).

We thank the reviewer for their thoughtful feedback.

1. Do the authors know that the polyubiquitin chains formed using the modified Ubiquiton strategy are only conjugated to the fused Ub moiety, but not the HMCES^{SRAP} domain? This was assumed, but need to be experimentally confirmed.

To identify lysine residues modified by the synthetic ubiquitylation system (Ubiquiton), we performed mass spectrometry analysis of HMCES^{SRAP} ubiquitylated by this system. HMCES^{FL}-DPCs modified by SUMO-targeted ubiquitylation served as control. Despite comparable overall coverage (84% vs. 70%), no major ubiquitylation was detected at any HMCES^{SRAP} lysine residue using the Ubiquiton system, whereas multiple HMCES lysine residues were modified by SUMO-targeted ubiquitylation (Fig. R6a-b). This result is in line with previous work showing that the Ubiquiton system targets specifically the fused-ubiquitin moiety (Renz et al, Mol Cell, 2024).

Fig. R6. (a) Mass spectrometry analysis of lysine residues within HMCES^{SRAP}-L20-Ub(G76V) after synthetic ubiquitylation (Ubiquiton) and HMCES^{FL} upon SUMO-targeted ubiquitylation. Violin blots show the mean \pm SD of three (Ubiquiton) and four (SUMO-targeted Ub-HMCES) biological replicates. (b) Peptide coverage of HMCES^{SRAP}-L20-Ub(G76V) and HMCES^{FL} used for ubiquitylation with the Ubiquiton system and SUMO-targeted ubiquitylation, respectively, measured by mass spectrometry. Only peptides with counts ≥ 10 (Ubiquiton) and ≥ 75 (SUMO-targeted ubiquitylation) are shown.

2. In the in vitro cleavage of DPC experiments, two major cleavage products with well-defined bands were detected. What is the molecular nature of these cleavage products. Knowing this is important considering that different mutants of SPRTN showed distinct effects on the formation of these two products.

We attempted to address this point using mass spectrometry. We digested unmodified and ubiquitylated HMCES^{SRAP}-DPCs following cleavage by SPRTN, using GluC with the hope to be able to identify semi-specific peptides (with one terminus cleaved by SPRTN and the second by GluC) using mass spectrometry (Fig. R7a-b). We compared the appearance and disappearance of semi-specific peptides in the presence of SPRTN. However, while we identified some semi-specific peptides, most were also detected in the absence of SPRTN or only upon cleavage of K48-modified but not K63-modified DPCs (or vice versa), making it unlikely that they represent the cleavage events observed in our gel-based assays.

Our efforts were complicated by the fact (i) that our reactions contain an excess of free, non-crosslinked HMCES^{SRAP}, and (ii) that cleaved peptide remnants remain attached to the DNA, making their detection by mass spectrometry more challenging. While we do consider this question interesting in principle, further optimization and experimentation will be required to map SPRTN cleavage sites. We will continue investigating this aspect but believe that a detailed mapping of the cleavage sites is beyond the scope of our study and is not essential to support the central conclusions of our current manuscript.

Fig. R7. (a) Reactions used for MS-based detection of proteolytic cleavage sites. Indicated HMCES^{SRAP}-DPCs (10 nM) were incubated alone or in the presence of FANCJ (100 nM), and SPRTN (100 nM) for 1 h at 30°C. (b) Peptide coverage of indicated HMCES^{SRAP}-DPCs either in the absence of SPRTN (Control) or upon cleavage by SPRTN (+SPRTN), referring to the experiment shown in (a), followed by GluC digest and mass spectrometry. Black dots mark sites cut by GluC, while unspecific cut sites are marked in red. Peptides only detected in samples containing SPRTN are marked with asterisks. Amino acid positions refer to HMCES^{SRAP}. Shown is summary data of three biological replicates.

3. Have the authors added free Ub to the assays described in Figure 1 d and e, particularly when unmodified HMCES^{SRAP} was used, to determine the effect on DPC cleavage? If the concentration of free Ub is high enough, a stimulation of SPRTN's protease activity is expected.

Free tetra-ubiquitin chains do indeed boost the cleavage of unmodified HMCES^{SRAP}-DPCs *in trans*. See response to Reviewer #1 – point 2 and New Extended Data Fig. 1b in the revised manuscript.

4. As seen in Figure 2m, the effect of Ub is rather small on SprT-BR with minimal chemical shift changes. However, DNA binding greatly enhanced the effect of Ub on SprT conformation. A model should be proposed to explain this interesting observation.

Indeed, the effect of ubiquitin on SPRTN is particularly pronounced in the presence of DNA. We discuss this observation more prominently in the revised version and propose a model in which DNA binding occurs first, triggering an open conformation that is then stabilized by ubiquitin binding to the USD interface.

5. As shown in Figure 3, the effect of L99S on DPC cleavage is less pronounced when the total DPC cleavage products are compared, while the amount of the smaller cleavage product DPC* was significantly reduced by the L99S mutation. This needs to be explained by knowing the molecular nature of the two DPC cleavage products.

Redacted

6. The DPC cleavage results obtained using DPC with either K48 or K63-linked polyUb chains showed little differences. This is a somewhat unexpected observation considering that the K48 chain is more compact, which may obstruct the binding of the Ile44 patch in the polyUb chain to the USD in the SPRTN's catalytic core domain.

Although K48-linked ubiquitin chains typically adopt compact conformations, they are dynamic in solution and can adopt open conformations with an exposed Ile44 patch (e.g. Lu et al, Structure, 2020). Our observation that both K48- and K63-linked ubiquitin chains activate SPRTN comparably, is entirely in line with a recent preprint by Kristijan Ramadan's group (Oxford, Singapore) which reports highly similar results (Song et al., bioRxiv, 2024).

7. In Figure 4h, the effect of L99S mutation on the cleavage of DPC was minimal in the presence of enzymatically installed polyubiquitin chains. This is different from the artificially installed polyUb chains. While it is known that the polyubiquitin chains are installed at different sites on HMCES, it is still difficult to rationalize this observation.

We do not entirely agree with this statement. In the synthetic system, all DPCs are modified, efficiently stimulating DPC cleavage through the two mechanisms, explained above. In contrast, SUMO-targeted ubiquitylation only results in a fraction of the DPCs being modified (See our response to Reviewer #1 – point 7). Nonetheless, SUMO-targeted ubiquitylation stimulates the formation of additional cleavage

fragments, which is reduced in experiments with the L99S mutant variant (Fig. 5h, compare lanes 5 and 14).

Reviewer #4 (Remarks to the Author)

This paper explores the relationship between DNA-protein crosslinks (DPC) ubiquitylation and SPRTN protease activation associated with their degradation by implementing a combination of experimental and computational techniques to understand the regulatory role played by ubiquitylation in SPRTN mediated DNA repair mechanisms. Results from in-vitro assays of ubiquitylated DPC's treated with a truncated SPRTN (SprT-BR), combined with structure prediction methods reveal a novel open conformation of SPRTN where a hydrophobic region (L38, L99) in SprT-BR interacts with ubiquitin. MD simulations and NMR experiments of SprT-BR mutants further confirm the role of this interactions, leading authors to propose a model in which the open conformation of SPRTN is stabilized by simultaneous binding of DNA and ubiquitin binding. Authors finally extend their results to investigate the role of ubiquitin-dependent activation of truncated SPRTN variants as seen in Ruijs-Aalfs (RJALFS) syndrome patients.

We appreciate the reviewer's perspective and support.

1. The authors novelty in this paper lies in the identification of a ubiquitin-binding interface at the SprT domain (USD). However, at the introduction they also mention a similar motif interacting with ubiquitin-binding region (MIU) which is not clearly explained in the paper or in the corresponding citation that mentions MIU (27). Clear distinction between MIU and USD would further strengthen the papers findings.

The MIU domain was predicted and discussed in the corresponding citation but never experimentally investigated. The fact that SPRTN truncations lacking the MIU are activated by DPC ubiquitylation to the same degree than the WT protein (Fig 1f), clearly demonstrate that the MIU is not involved.

2. For the computational methodology, authors should mention how h++ provides an estimation of the protonation states of titratable residues.

We thank the reviewer for the comment and included additional information in the methods part of the revised manuscript (page 23-25) to address this point. The pdb-files (SprT-WT.pdb; SprT-L38S.pdb; SprT-L99S.pdb) with the protonation state of all residues are now provided as supplementary data.

3. Authors should also specify the number of clusters and exact criteria for such for computational reproducibility, such as the RMSD of all atoms or backbone CA atoms.

To improve clarity and reproducibility, we added more information on the clustering algorithm used and on the employed cutoff in the revised methods section for MD simulations (page 23-25). We summarized the numbers of clusters and the fraction of structures in the three most populated clusters in Supplementary Table S2.

4. MD simulations of SPRT mutants can highlight experimental highlights further and provide molecular insights into the disruptive effects of L38S.

Fig. 2j-l in the main manuscript illustrates the mutation sites and their effects on binding affinity. However, we agree that further analysis could better highlight the molecular impact of SprT mutations.

Initially, we identified the changes in the interaction patterns by calculating the minimum distances of all amino acids in ubiquitin to the residues at positions Leu38 (Fig. 2j) and Leu99 (Fig. 2k) for the wild-type and the two mutated structures and performed alanine scanning to determine the binding energy for SprT-Ub¹ in SprT-L38S and SprT-L99S (Fig. 2l).

Fig. 2. Ubiquitin stabilizes an open SPRTN conformation. (i) Main MD-cluster of the indicated structure during MD simulation for 400 ns, generated from three independent trajectories. For SprT (ColabFold predicted) two of three main MD-clusters are depicted. R_g correlating frequencies among all performed simulations are labeled above. (j-k) Zoom-in to regions i and ii of the SprT-Ub1 complex (i), showing amino acids of ubiquitin (in grey) surrounding residue Leu38 (j) or L99 (k) of SPRTN (in blue) in the wild-type (WT) protein (left) and upon L38S or L99S replacement, respectively (right). (l) SprT+Ub1 binding energy difference ($\Delta\Delta G$) between SprT-L38S or -L99S and WT protein obtained from alanine scanning. Bar graphs show the mean \pm SD of 301 snapshots from PBSA calculations for the central structure of the largest cluster.

In addition, we calculated the average number of contacts for SPRTN's residues Leu38 and Leu99 with ubiquitin for the wild-type protein and the USD variants L38S and L99S (New Extended Data Fig. 2e). Hereby, we define a contact as a pair of atoms with less than 0.6 nm distance. We observe that the L38S mutation reduces the number of contacts at position Leu38, while leaving position Leu99 largely unaffected. Conversely, the mutation at position Leu99 decreases contacts at that site but has minimal impact on position Leu38 (New Extended Data Fig. 2e). These observations support the notion that the L38S and L99S mutation disrupt contacts between SPRTN and ubiquitin at the interface. We conclude that the replacement of Leu38 or Leu99 leads to a loss of hydrophobic contacts to ubiquitin's Ile44- and Ile36-patch (New Extended Data Fig. 2d-e) and hence to a decrease in binding affinity.

In addition to the interaction pattern, we included the bar chart showing the number of contacts of SprT with ubiquitin in the revised manuscript (New Extended Data Fig. 2d-e).

New Extended Data Fig. 2. A novel ubiquitin binding interface at the SprT domain. (d) Heat map indicating minimum distances of all amino acids in ubiquitin to residues at positions 38 (upper part) and 99 (lower part) of SPRTN, including wild-type (WT) conditions (L38, L99) and mutated states (L38S, L99S) (top). Structure of

ubiquitin colored by hydrophobicity (bottom). Ile36 and Ile44 patches are highlighted. (e) Average number of contacts between SprT residues L38 and L99 and ubiquitin (Ub). Contacts were defined as any interatomic distance smaller than 0.6 nm between side chain or backbone atoms of the respective SprT residue (either L38 or L99) and any atom of ubiquitin. For comparability, contacts for multiple atoms of the SprT residue with an atom of ubiquitin are only counted once. For the WT protein, residue L38 forms approximately 60 contacts with ubiquitin, while residue L99 forms around 130 contacts. Upon mutation of L38 (L38S), the number of contacts at that position is reduced to around 20, while the number of contacts at L99 remains largely unchanged. Conversely, in the mutant where L99 is altered (L99S), the contacts at residue L99 decrease to around 68, while the interaction at L38 remains similar to the WT.

Note:

In addition to the changes based on the Reviewer's comments, we optimized buffer and measurement conditions for our NMR experiments. As a result, we have replaced all previously included NMR spectra with newly acquired data of improved quality. The conclusions drawn from these new measurements remain entirely unchanged.

REVIEWERS' COMMENTS

Reviewer #1 (Remarks to the Author):

The authors have clarified all the points I raised during the review by either adding additional data or by discussion. The revised manuscript is of a high quality and I recommend acceptance.

Reviewer #2 (Remarks to the Author):

Manuscript by Durauer et al. is a comprehensive biochemical study delineating the role of ubiquitin in SPRTN activation and DPC proteolysis. The authors have successfully addressed the reviewers queries. The Identification of a ubiquitin-binding interface at the SprT domain (USD) in SPRTN, and the role of ubiquitin in SPRTN-mediated DPC proteolysis is significant. The revised manuscript is suitable for publication.

Reviewer #3 (Remarks to the Author):

The authors have addressed the reviewer's questions satisfactorily.

Reviewer #4 (Remarks to the Author):

Thank you for taking into account the previous suggestions.

RESPONSE

We thank the reviewers for their positive assessment and support.